# Competing processes determine the long-term impact of basal friction parameterizations for Antarctic mass loss

Tim van den Akker[1], William H. Lipscomb[2], Gunter R. Leguy[2], Willem Jan van de Berg[1], Roderik S.W. van de Wal[1,3]

[1]Institute for Marine and Atmospheric Research Utrecht, Utrecht University, Netherlands

[2]Climate and Global Dynamics Laboratory, NSF National Center for Atmospheric Research, Boulder, CO, USA

[3]Department of Physical Geography, Utrecht University, Netherlands

*Correspondence to*: Tim van den Akker (t.vandenakker@uu.nl)

**Abstract.** Previous studies do not agree on the magnitude of the influence of basal friction laws in sea-level projections. We use the Community Ice Sheet Model (CISM) to show that the sensitivity of the projected sea level rise to the choice of basal friction law depends on the specific geometric setting and the initial state of the ice sheet model. We find a geometry-driven connection between buttressing and basal sliding in the Amundsen Sea Embayment, when doing unforced (i.e. based on the present day observed imbalance of the Antarctic Ice Sheet) future simulations of several centuries in which Thwaites Glacier and Pine Island Glacier eventually collapse. We perform two initializations: one were we tune a free parameter in the basal friction parameterization and the ocean temperatures based on ice thickness misfits, and one in which we additionally tune a parameter in the ice viscosity calculation based on ice surface velocity misfits. Both initialization lead to a modelled Antarctic Ice Sheet that well resembles present-day observed conditions (ice thickness, ice surface velocities and mass changes rates). In the simulations following the first initialization, Thwaites Glacier collapses first. In the simulations following the second initialization, Pine Island Glacier collapses first. When Thwaites Glacier collapses first, it creates a grounding line flux large enough to sustain an ice shelf that provides buttressing which largely balances the basal friction differences when using different basal friction parameterizations. Such an ice shelf does not form, however, when Pine Island Glacier retreats significantly, because it deglaciates slower with typically a lower grounding line flux, so the basal melt parameterization is able to reduce the amount of buttressing of the newly formed shelf. Consequently, a collapsing Pine Island glacier is sensitive to the choice of basal friction law, but a collapsing Thwaites Glacier is potentially not. Which glacier collapses first depends on the initial state, in our case the inversion procedure. The sparsity of observations in combination with the considerable amount of parameterization and arbitrary choices in our ice sheet modelling allow that the sensitivity to basal friction of the collapse for PIG and Thwaites can be large or small depending on the details of the initialization.

# 1 Introduction

The projected Antarctic contribution to global mean sea level (GMSL) rise ranges from 0.03 - 0.27 m (SSP1.9) to 0.03 - 0.34 m (SSP8.5) in 2100 (Fox-Kemper et al., 2021). After 2100, uncertainty increases because of dynamical processes, leading to a possible multi-meter GMSL rise by 2300 (Fox-Kemper et al., 2021; Payne et al., 2021; Seroussi et al., 2024). By and after 2300, self-sustaining processes could cause the deglaciation of large parts of the West Antarctic Ice Sheet (WAIS) (Joughin et al., 2014; Cornford et al., 2015; Seroussi et al., 2017; Coulon et al., 2024; Van Den Akker et al., 2025).

The largest dynamic mass changes are currently ongoing in the Amundsen Sea Embayment (ASE) (Smith et al., 2020). The possible deglaciation of the two largest glaciers in this region, Pine Island Glacier (PIG) and the Thwaites Glacier (TG), is therefore one of the main sources of uncertainty in modelled ice sheet future behavior (Cornford et al., 2015; Feldmann and Levermann, 2015; Arthern and Williams, 2017; Pattyn and Morlighem, 2020; Bett et al., 2023; Seroussi et al., 2024). The grounding line of both glaciers rests on a retrograde bed, making them susceptible to the Marine Ice Sheet Instability (MISI, see Schoof (2007b) ). Some studies (Joughin et al., 2014; Favier et al., 2014) suggest that those glaciers are already undergoing MISI-like retreat. Recent studies suggested that present-day ocean temperatures could drive complete deglaciation of this area over several centuries, without additional warming (Reese et al., 2023; Van Den Akker et al., 2025).

Sources of modelled ice sheet uncertainty could be missing representation of certain physical processes, or a suboptimal initial state (Aschwanden et al., 2021). Several studies have attributed uncertainty in sea level prediction from ice sheet models to the choice of the basal friction parameterization (Brondex et al., 2017; Bulthuis et al., 2019; Brondex et al., 2019; Sun et al., 2020; Wernecke et al., 2022; Barnes and Gudmundsson, 2022; Berdahl et al., 2023; Joughin et al., 2024). These parameterizations are relations between ice basal velocities and the friction at the ice-bedrock interface. Generally, the existing relations, or basal friction laws, can be separated into two categories. The first category is a relation where friction depends on the basal velocity raised to some power. Using an exponent of 1 results in a linear relation, but exponents between 0 and 1 are more common (Weertman, 1957; Budd et al., 1979; Barnes and Gudmundsson, 2022; Das et al., 2023). These are referred to as 'power law friction', and were originally developed to represent basal sliding over hard bedrock. The second category is a relation in which the friction becomes independent of velocity for fast- flowing ice. This is referred to as 'Coulomb friction' (Schoof, 2005; Tsai et al., 2015; Joughin et al., 2019; Zoet and Iverson, 2020), and this friction law was originally developed to represent sliding over softer, deformable till. Both types of basal friction laws usually contain free parameters that can be tuned to match observed quantities such as ice sheet surface velocities or thickness.

Another source of uncertainty is the potential of ice shelves to provide a buttressing force on the inland ice sheet (Dupont and Alley, 2005; Gudmundsson, 2013; Fürst et al., 2016; Haseloff and Sergienko, 2018; Reese et al., 2018a). A buttressed ice shelf can act as plug against glacier acceleration. An accelerating glacier has an increasing grounding line flux, transporting more ice to the ice shelf. If the thicker shelf can persist, this will increase its buttressing capacity and oppose the initial acceleration

of the inland ice. Choices related to basal friction will influence both the velocity profile and the modelled buttressing of
simulated ice sheet and ice shelves.

Most of the literature argues that power law friction will result in less modelled sea level rise compared to Coulomb friction, both for idealized experiments (e.g. MISMIP-style experiments, see Asay-Davis et al. (2016)) and for realistic simulations of the Antarctic Ice Sheet (AIS). However, authors do not agree if the difference is substantial (Brondex et al., 2017; Sun et al.,
2020; Brondex et al., 2019) or not (e.g., (Barnes and Gudmundsson, 2022; Wernecke et al., 2022). Furthermore, basal friction parameterization tests in realistic settings such as the ASE or the entire AIS are often done with enormous (ABUMIP; (Sun et al., 2020) or unrealistic perturbations (Barnes and Gudmundsson, 2022; Brondex et al., 2019). Tests with realistic forcing and therefore realistic mass change rates, as well as a more generic study of the importance of the choice of the friction law including its parameter is missing in studies of a potentially realistic, slow WAIS collapse.


Here we use the Community Ice Sheet Model (CISM) (Lipscomb et al., 2019; Lipscomb et al., 2021) to investigate the sensitivity of grounding line retreat and ice mass loss to the choice of basal friction laws for the West Antarctic Ice Sheet for realistic collapse conditions. Building on the work of Brondex et al. (2017) and Brondex et al. (2019), we extend our simulations for 2000 years into the future, to capture the importance of significant grounding line retreat. We use the observed
mass change rates from Smith et al. (2020) to initialize the AIS into the observed state of regional mass loss, applied in the same way as in Van Den Akker et al. (2025): by including the mass change rates into the ice transport equation during the initialization period. We perform two initializations: in both initializations we tune ocean temperature perturbations to obtain a match between modelled and observed ice shelf thickness. In one of the initializations we furthermore only tune the free parameter in the basal friction laws to nudge the modelled ice sheet thickness toward observations, and in the second
initialization we additionally tune a flow enhancement factor to nudge the modelled ice surface velocities closer to observations. We then continue our simulations with four widely used friction laws. We present a specific case in which a reduction in basal friction, caused by adopting a different basal sliding law, is offset by a corresponding increase in buttressing. Whether this compensation occurs ultimately depends on the glacier's geometric evolution. Some geometry evolutions allow for compensating effects between buttressing and basal friction. The evolution itself is controlled by specific choices made
during the initialization phase. Hence, the intitialization can determine wheter the modelled ice sheet evolution is sensitive to the choice of basal friction law or not.

In Section 2 we describe the basal friction laws used and two different ways of calculating the buttressing capacity, as well as the ice sheet model CISM. In Section 3 we show the specific geometric setting of TG, PIG and the ASE. In Section 4 both the
results of our two inversions and the continuation experiments are shown. A discussion is presented in Section 5, and this study ends with conclusions presented in Section 6.

## 2 Methods

### 2.1 Basal friction

We test four basal friction laws. First, we use a Regularized Coulomb sliding law proposed by Zoet and Iverson (2020),
hereafter referred to as 'Zoet-Iverson law', representing regularized Coulomb friction:

$$\tau_{b,Reg} = C_c(x,y)N(x,y)\left(\frac{u_b}{u_b+u_0}\right)^{\frac{1}{m}} \tag{1.1}$$

where $\tau_{b,Reg}$ is the basal friction is $N$ the effective pressure, $C_c$ is a spatially varying unitless tuning parameter in the range
[0,1] controlling the strength of the Coulomb sliding, $u_0$ is the yield velocity and $m$ a modeler-defined exponent, chosen to
be 3 as commonly done. For their short description and units, see Table S1 and S2. The spatial-varying parameter $C_c$
corresponds to the $\tan\phi$-term of Zoet and Iverson (2020), Eq. 3, in which $\phi$ is the friction angle, a material property of the
subglacial till. This parameter is used to nudge our modelled ice sheet toward the observed ice thickness by a process described
in the next section.

In addition to the Zoet-Iverson friction law, we consider three more relations:

$$\tau_{b,Powerlaw} = C_p(x,y) * u_b^{\frac{1}{m}} \tag{1.2}$$

$$\tau_{b,Schoof} = \frac{C_p C_c N}{\left[C_p^m u_b + (C_c N)^m\right]^{\frac{1}{m}}} u_b^{\frac{1}{m}} \tag{1.3}$$

$$\tau_{b,Pseudoplastic} = C_c N\left(\frac{u_b}{u_0}\right)^{\frac{1}{pp}} \tag{1.4}$$

In these equations, $C_p$ is a spatially varying constant. Eq. 1.2 is the classical power law for sliding ('power law' hereafter)
from Weertman (1957). Note that we use for $m$ the same value as in Eq 1.1; it regulates the strength of the power law friction.
Eq 1.3, often referred to as the "Schoof law" (Schoof, 2005), is a regularized Coulomb friction law suggested for the Marine
Ice Sheet Model Intercomparison Project third phase (MISMIP+) experiments (Asay-Davis et al., 2016) and used in CISM in
Lipscomb et al. (2021). It is argued in the literature that Eq 1.3 is preferred over Eq 1.2 because it yields physically realistic
behavior of a retreating glacier (Brondex et al., 2017; Brondex et al., 2019). Eq 1.4 is referred to as a 'pseudoplastic' law,
developed and used by Winkelmann et al. (2011); Aschwanden et al. (2016). It is also often referred to as the 'Budd' law after

Budd et al. (1979). This last sliding law is similar in behavior to the power law but includes the effective pressure $N$, as in Eq. 1.3 and a value for $pp$ between 1 and 5.

The effective pressure $N$ at the ice–bed interface is the difference between the ice overburden pressure and the subglacial water pressure. In our simulations, the effective pressure is lowered near grounding lines to represent the connection of the subglacial hydrological system to the ocean (Leguy et al., 2014):

$$N = \rho_i gH \left(1 - \frac{H_f}{H}\right)^p \tag{1.5}$$

Where $\rho_i$ is the density of glacial ice, $g$ the gravitational acceleration, $H$ the ice thickness and $p$ is a constant in the range [0,1]. The flotation thickness $H_f$ is the height of an ice column resting on bedrock below sea level (b < 0) at hydrostatic equilibrium, which is given by:

$$H_f = \max\left(0, -\frac{\rho_w}{\rho_i} b\right) \tag{1.6}$$

where $\rho_w$ is the sea water density, and b is the bedrock height. Simulations in this study were done with $p = 0.5$ unless stated otherwise. At the grounding line, we apply friction and/or basal melt scaled by the percentage of the modelled grid cell that is grounded as in Leguy et al. (2021).

The four friction laws differ in their dependence on the ice basal velocities and are shown with typical values in Fig. 1. The power law and the pseudoplastic law differ clearly from the Schoof and Zoet-Iverson law as they do not approach a limit for high basal velocity. The Schoof and Zoet-Iverson law behave similarly, although the Zoet-Iverson law is slightly more sensitive to changes in velocity. The power law and pseudoplastic law show increasing basal friction with increasing velocity with diminishing impact. This introduces a negative feedback: a decrease in friction initially increases velocity, which in the power law and pseudoplastic law will always increase the friction. In the other two laws, the basal friction asymptotes for high velocities to $C_c N$, which is pure Coulomb friction. This has the important implications that perturbations introduced to simulations using power law or pseudoplastic friction will be damped quicker and more locally. A sudden loss of buttressing leading to a local speed up of ice at the grounding line (where ice velocities often exceed 200 m yr[-1]) will be 'braked' more heavily when using the power law, preventing the perturbation from influencing ice (far) upstream of the grounding line.

Three of the sliding laws depend on the effective pressure. This introduces a feedback between ice velocities and basal friction. If the basal friction decreases, the ice flux across the grounding line increases. This decreases the ice thickness

upstream of the grounding line, further reducing $N$. Thinning grounded ice, for example caused by a loss of buttressing, can

lower $N$ and lead to thinning far upstream of the grounding line (Brondex et al., 2017; Brondex et al., 2019).

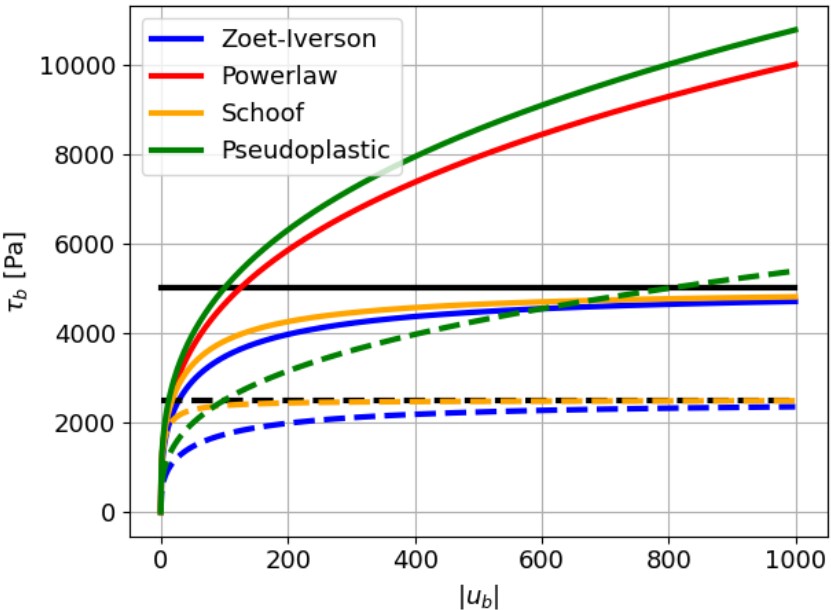

**Figure 1. The four friction laws with indicative values.** Basal friction as function of the ice basal velocity for constant values of $C_c = 0.5$, $C_p = 1000$, $N = 10000$ Pa, $u_0 = 250$ m/yr, and $m = 3$. Dashed lines are the same basal friction laws with N=5000 Pa (black line). The

power law is independent on N, so the solid and dashed red line are the same. Coulomb sliding asymptotes (e.g. the product $C_c N$) for the dashed and solid lines.

## 2.2 Buttressing

We quantify the buttressing capacity of an ice shelf at the grounding line with two approaches. The first approach compares the stress balance at the grounding line to the stress boundary condition at the calving front. This buttressing number, defined

as the ratio between the latter two stresses, has been used in several studies (Gudmundsson, 2013; Gudmundsson et al., 2023; Fürst et al., 2016) and as a parameter in the analytical grounding line flux of Schoof (2007a) and Schoof (2007b). Reese et al. (2018a) describe three ways to calculate the buttressing number, of which we choose Eq. 11 in their paper and adapt it according to Eq. S1 in the supplementary material of Fürst et al. (2016):

$$\chi = 1 - \frac{\boldsymbol{n_1}.\boldsymbol{R}\boldsymbol{n_1}}{2R_f} \tag{1.7}$$

The vector $\boldsymbol{n_1}$ is the vector perpendicular to the grounding line, in our regular rectangular grid best approximated by the

direction of the ice flow. The stress boundary condition at the 'would be' (i.e. if there would be no shelf and hence no buttressing) calving front $R_f$ is given by

$$R_f = \frac{\rho_i g}{4}\left(1 - \frac{\rho_i}{\rho_w}\right)H$$

(1.8)

The parameter $R_f$ therefore is the stress boundary condition if there would be a calving front at that position. The resistive stress tensor $\boldsymbol{R}$ is given by

$$\boldsymbol{R} = \begin{pmatrix} 2\tau_{xx} + \tau_{yy} & \tau_{xy} \\ \tau_{xy} & \tau_{xx} + 2\tau_{yy} \end{pmatrix}$$

(1.9)


The variable $\chi$ in Eq 1.7 has a value of 0 for areas that are unbuttressed: then the buttressing force equals the driving stress if the ice sheet ended at that point with an ice cliff. Values above zero indicate buttressing. Values below zero point to a tensile regime where the ice shelf is pulling grounded ice over the grounding line. As shown by Reese et al. (2018a), the buttressing number from the linearized stress balance approach depends on the choice of $\boldsymbol{n_1}$ and $\boldsymbol{R}$, and assumes that the stress tensor at the grounding line is determined by the buttressing capacity of the downstream shelf only.

The second approach to quantify the buttressing is by performing so-called shelf-removal experiments (e.g. Antarctic BUttressing Model Intercomparison Project (ABUMIP), Sun et al. (2020). In these experiments, floating ice is instantly removed, and the effect on the grounded ice in terms of acceleration is used to measure the buttressing capacity of the removed shelves. We define the acceleration number, in analogy to the definition of the buttressing number, by

$$\alpha = 1 - \frac{u_{before}}{u_{after}}$$

(1.10)

in which $u_{before}$ and $u_{after}$ refer to the local depth-averaged ice velocity before and after removing the shelf. This method of quantifying the buttressing is the simplest way of assessing shelf strength but provides only a temporal snapshot and requires an additional ice sheet model timestep to be calculated.


Both methods shown in Eqs 1.7 and 1.10 are tested on a theoretical case (the Ice1r experiment of MISMIP+, see Asay-Davis et al. (2016) and on the present-day state of the Antarctic Ice Sheet, before using them in the continuation simulations in this study. These results for the buttressing can be found in the supplementary material, Fig S1-S3.

## 2.3 The Community Ice Sheet Model

The Community Ice Sheet Model is a thermo-mechanical higher-order ice sheet model, which is part of the Community Earth System Model version 2 (CESM2, Danabasoglu et al. (2020). Earlier applications of CISM to the Antarctic Ice Sheet retreat can be found in Seroussi et al. (2020); Lipscomb et al. (2021); Berdahl et al. (2023); Van Den Akker et al. (2025). The variables

and constants used in the text and equations below are listed in Tables S1 and S2. All simulations in this study are done on a 4-km grid.


We run CISM with a vertically integrated higher-order approximation to the momentum balance, the Depth Integrated Viscosity Approximation (DIVA) (Goldberg, 2011; Lipscomb et al., 2019; Robinson et al., 2022). The momentum balance in the x-direction (the y-direction is analogue) is defined as:

$$\frac{\partial}{\partial x}\left(2\bar{\eta}H\left(2\frac{\partial\bar{u}}{\partial x}+\frac{\partial\bar{v}}{\partial y}\right)\right)+\frac{\partial}{\partial y}\left(\bar{\eta}H\left(\frac{\partial\bar{u}}{\partial y}+\frac{\partial\bar{v}}{\partial x}\right)\right)-\beta u_{x,b}=\rho_i gH\frac{\partial s}{\partial x} \tag{1.11}$$

Barred variables are depth averaged. Basal friction, which is parameterized in the ways described in Sec. 2.1, appears as the product of $\beta$ and the directional velocity in Eqs 1.11.

Since the spatially varying parameters $C_c$ and $C_p$ in Eq 1.1 – 1.4 are poorly constrained by theory and observations, we use it as a spatially variable tuning parameter. We tune $C_c$ using a nudging method (Lipscomb et al., 2021; Pollard and Deconto,
200 2012):

$$\frac{dC_c}{dt}=-C_c\left[\left(\frac{H-H_{obs}}{H_0\kappa}\right)+\frac{2}{H_0}\frac{dH}{dt}-\frac{r}{\kappa}ln\frac{C_c}{C_r}\right] \tag{1.12}$$

Where $\kappa$ is the relaxation timescale, and r a parameter controlling the strength of the relaxation term. A higher value for r will 'pull' the inverted $C_c$ towards $C_r$. In the end-member case, where r is set to infinite, $C_c$ and $C_r$ are equal. The relaxation target $C_r$ is a 2D field based on elevation, with lower values at low elevation where soft marine sediments are likely more prevalent, loosely following Winkelmann et al. (2011). We chose targets of 0.1 for bedrock below -700 m asl and 0.4 for 700
m asl, with linearly interpolation in between, based on Aschwanden et al. (2013).

Basal melt rates are calculated using a local quadratic relation with a thermal forcing observational dataset (Jourdain et al., 2020):

$$bmlt=\gamma_0\left(\frac{\rho_w c_{pw}}{\rho_i L_f}\right)^2(\max[TF_{base}+\delta T,0])^2 \tag{1.13}$$

where $TF_{base}$ is the ocean thermal forcing (the difference between the ocean temperature the local melting point) from Jourdain
et al. (2020), interpolated to the modelled ice shelf base. The ocean temperatures are tuned in order for the floating ice to match the thickness observations of Morlighem et al. (2020), similar to Eq. 1.12 but with $\delta T$ as tuning variable:

$$\frac{d(\delta T)}{dt} = -\delta T \left[ \left( \frac{H - H_{obs}}{H_0 \kappa} \right) + \frac{2}{H_0} \frac{dH}{dt} \right] + \frac{(T_r - \delta T)}{\kappa} \tag{1.14}$$

As in Eq 1.12, we add a term including a relaxation target $T_r$ to penalize large deviations. In this case, the relaxation target is zero, since $\delta T$ is a temperature correction to the dataset of Jourdain et al. (2020). The melt sensitivity $\gamma_0$ is chosen to be $3.0 \times 10^4$ m/yr, which was used in Lipscomb et al. (2021) and Van Den Akker et al. (2025) to obtain basal melt rates in good agreement with observations and with a shelf-average $\delta T$ close to zero in the Amundsen Sea Embayment.

Additionally, a flow enhancement multiplication factor $E$ can be tuned to nudge modelled ice surface velocities towards observations in a similar way:

$$\frac{dE}{dt} = E \left[ -\left( \frac{u_s - u_{s,obs}}{v_0 \kappa} \right) + \frac{2}{H_0} \frac{dH}{dt} + r \, log \frac{E}{E_r} \right], \tag{1.15}$$

in which $v_0$ is velocity scale of the inversion and $E_r$ the relaxation target. Since $E$ is a multiplication factor, a value of 1 equals no enhancement, and $E_r$ is also set to1. Following Lipscomb et al. (2021); Van Den Akker et al. (2025), a value of 1 is used for grounding ice and 0.5 for floating ice if not inverting for $E$. These values are also the initial values used when starting an inversion using Eq 1.15. The flow enhancement factor is then used in the calculation of the rate factor $A$:

$$A = EA_0 e^{-Q/(RT^*)}, \tag{1.16}$$

with $T^*$ as the homologous temperature and $A_0, Q, R$ as constants (see Table 2):

$$T^* = T + \rho g H \Phi. \tag{1.17}$$

Then, A is used together with the strain rates $\epsilon$ to calculate the effective viscosity of the ice, $\eta$, which controls the ice velocity:

$$\eta = \frac{1}{2} A^{-\frac{1}{n}} \epsilon_e^{\frac{1-n}{n}}. \tag{1.18}$$

The effective strain rate $\epsilon_e$ is defined as the norm of the strain-rate tensor:

$$\epsilon_e = \frac{1}{2} \epsilon_{i,j} \epsilon_{i,j}, \tag{1.19}$$

with

$$\epsilon_{i,j} = \frac{1}{2} \left( \frac{\partial u_i}{\partial x_j} + \frac{\partial u_j}{\partial x_i} \right). \tag{1.20}$$

We tune for both the basal friction coefficient and the flow enhancement factor in this study. We define 'inversion' here as the process of retrieving, tuning or nudging a parameter (i.e. the flow enhancement factor) from observables (ice surface velocities). This flow enhancement factor inversion can interfere with the basal friction inversion in Eq. 1.12. For example, if the modelled ice is thinner than observed in a grid cell, the basal friction inversion tends to increase $C_c$ via Eq. 1.12 to slow down and pile up the ice at that location. By doing this, the surface velocity decreases. If the modelled surface velocities fall below the observed velocities, the flow enhancement factor will try to increase to speed up the ice, counteracting the ice thickness increase caused by the basal friction inversion. This creates a conflict between the two inversion, namely when modelled ice is both too slow and too thin, or vice versa, too fast and too thick:

$$\frac{dC_c}{dt} > 0 \ \wedge \ \frac{dE}{dt} < 0 \ , \tag{1.21a}$$

$$\frac{dC_c}{dt} < 0 \ \wedge \ \frac{dE}{dt} > 0 \ . \tag{1.21b}$$

The basal friction inversion is the primary and default inversion. It has been successfully applied to the AIS in an ISMIP6-style (Ice Sheet Model Intercomparison Project for CMIP6, Nowicki et al. (2016)) setting in Lipscomb et al. (2021) and Van Den Akker et al. (2025) , while the flow enhancement factor inversion has been developed for this study, and is therefore less tested and validated. For this reason, whenever there is a conflict between the flow enhancement factor inversion and the basal friction inversion, We allow $C_c$ to be changed, but set $\frac{dE}{dt}$ to zero at that timestep and location. Finally, to prevent overfitting, the flow enhancement factor is only allowed to be changed in grid cells where the ice surface velocity mismatch is larger than 25 m yr$^{-1}$.

The grounding line (GL) is not explicitly modeled in CISM, but its location can be diagnosed from the hydrostatic balance. Since we use a regular rectangular grid, the modeled GL cuts through cells. To prevent abrupt jumps in the basal friction and the basal melt rates close to the GL, we use a GL parameterization (Leguy et al., 2021), where we use a flotation function to weigh the basal friction and basal melt rates according to the percentage of a grid cell that is grounded.

## 2.4 Initializations

In this study, we use two initializations. The 'default initialization' (abbreviated as 'DI') uses the observed ice thickness as a target, nudges free parameters in the basal friction (Eq. 1.12) and basal melt parameterizations (Eq. 1.14), and per construct starts a continuation run with the observed mass change rates from Smith et al. (2020). The second initialization, the 'Flow Enhancement Factor initialization' (abbreviated as 'FEFI'), additionally targets observed ice surface velocities by nudging the flow enhancement factor $E$ (Eq 1.15). The former initialization is used in other CISM applications to the AIS (Lipscomb et al.,

2021; Berdahl et al., 2023; Van Den Akker et al., 2025). Other studies, which use the 'Data Assimilation' method to initialize their ice sheet models, include by default the observed ice surface velocities as targets (Bradley and Arthern, 2021; Larour et al., 2012; Cornford et al., 2015; Arthern et al., 2015). Both initializations are tested on their stability. We deem an initialization successful and 'stable' when there is little to no instantaneous model drift once we turn of the inversions and keep our nudged parameters constant, and no significant changes in modelled ice sheet geometry when run forward for 2000 years, similar to what was done by Van Den Akker et al. (2025)

### 2.4.1 Default initialization (DI)

The default initialization uses Eq 1.12 and Eq 1.14 to initialize an Antarctic Ice sheet in equilibrium, with ice thicknesses approximately matching the observations of Morlighem et al. (2020) and observed thinning/thickening rates ($dH/dt$) from Smith et al. (2020) (see section 4.1 and Figures 4 and 5). At the end of the inversion process, the resulting ice velocities are in good agreement with the observed surface velocities from Rignot et al. (2011). The observed $dH/dt$ is imposed as an additional term in the mass transport equation during the initialization, as described in Van Den Akker et al. (2025). We start the initialization by providing CISM the observed ice thickness from Bedmachine version 1 from Morlighem et al. (2020) after which the thickness is allowed to evolve. At every following timestep, CISM nudges the free parameter in the friction law ($C_c$ in Eq 1.1) and the ocean temperature correction ($\delta T$ in Eq 1.13) to decrease the modelled thickness mismatch with the observations. A successful initialization is considered complete when the modelled ice sheet thickness converges; the resulting modelled thickness, surface velocities, grounding line position and basal melt fluxes are close to their observed values; and forward simulations with continued imposed $dH/dt$ display minimal drift. For normal forward simulations, the observed $dH/dt$ is no longer added to the mass transport equation, so that these simulations start with thinning rates equal to the observed thinning rates, as in Van Den Akker et al. (2025).

### 2.4.2 Flow Enhancement Factor Initialization (FEFI)

Often in ice sheet models, the effective viscosity depends on the strain rates and a flow factor, with the flow factor dependent on the ice temperature via an Arrhenius relation. However, impurities and other factors, e.g., damage, fabric formation and (local) anisotropy and errors in the temperature field, may also influence the flow factor, hence an enhancement factor $E$ can be introduced to scale the flow factor where necessary.

If $E$ increases, the effective viscosity decreases (Eq. 1.16), which will increase the ice deformation and the ice surface velocities. This inversion changes the vertical structure of the velocity profile, altering the difference between the surface and basal velocities. Hence, changes in $E$ can make a region more deformation-dominated (as in the Shallow Ice Approximation, or SIA) or sliding-dominated (as in the Shallow Shelf Approximation, or SSA).

We use the three observational datasets (ice thickness, ice surface velocities, and the mass change rates) listed above as nudging targets; each has different timestamps, resolutions, and uncertainties. We use two (DI) or three (FEFI) free parameters to nudge the modelled present-day ice sheet towards observations. In both cases, the system is underdetermined, i.e. multiple combinations of tuned free parameter values will result in a very similar modelled ice sheet. Moreover, the nudging procedure has two degrees of freedom in the FEFI simulations to match observations of observed ice thickness and ice surface velocities.

Due to the inconsistent datasets, this might lead to a well-matching thickness thanks to the basal friction inversion, a well-matching ice surface velocity thanks to the FEFI, but physically implausible behavior, e.g., deformation-driven flow in fast-flowing regions close to the grounding lines where sliding-dominated flow is expected. The nudging procedure is free, for example, to create sliding or deformation regimes where mathematically preferred, to minimize the difference between observed and modelled surface velocities. The FEFI simulation might therefore have greater skill in representing observations,

but for the wrong reasons. Our goal in this study is not directly to simulate the AIS as realistically as possible, but rather to show the interaction of basal friction laws and the ice shelf geometries during deglaciation phases of the WAIS.

## 2.5 Continuation experiments

    We carry out continuation experiments to test the modelled ice sheet evolution sensitivity to the choice of basal friction law,

starting from the two initializations described above. We do not apply any further climate forcing, so oceanic and atmospheric temperatures as well as the surface mass balance are kept constant in time.  Each run consists of either 1000 or 2000 model years, depending on the WAIS deglaciated state after 1000 years or not. We focus on the WAIS due to its greatest dynamic changes in both the observations (Smith et al., 2020) and modelled ice sheet.

The initial state of the modelled AIS will differ slightly if we perform initializations with different friction laws. We would like to start our experiments from the same initial state. Therefore, we rewrite free parameters in the friction laws, $C_p$ and $C_c$, to allow us to start continuation runs with different basal sliding laws from the same initialized state as was done by Brondex et al. (2017) and Brondex et al. (2019). This has two advantages. First, there are initially no differences in geometry between continuation runs, so arising differences during the continuation experiment can be attributed to the choice of the basal sliding

law. Second, the initialization typically takes about 10,000 model years, while a continuation only requires 1000–2000 yr, saving computational expenses. We take the initializations using the ZI law as initial states. The details of rewriting the free parameters are described in the supplementary material.

### 3. Amundsen Sea Embayment

The Amundsen Sea Embayment is presented in Fig.2. Both PIG and TG are flanked by bedrock above sea level and separated by a small ridge that is well below sea level but has some prominence compared to the troughs on both sides (Fig. 3). The basin boundary (as defined by Zwally et al. (2015) crosses over this ridge. The present-day grounding line is situated at a chokepoint: as it recedes upstream into the troughs, the distance between the left flank of PIG and right flank of TG becomes larger. However, if the grounding line recedes, ice shelves can remain in place for both PIG and TG, locked at the narrow point

between the two flanks where the grounding line currently exists.

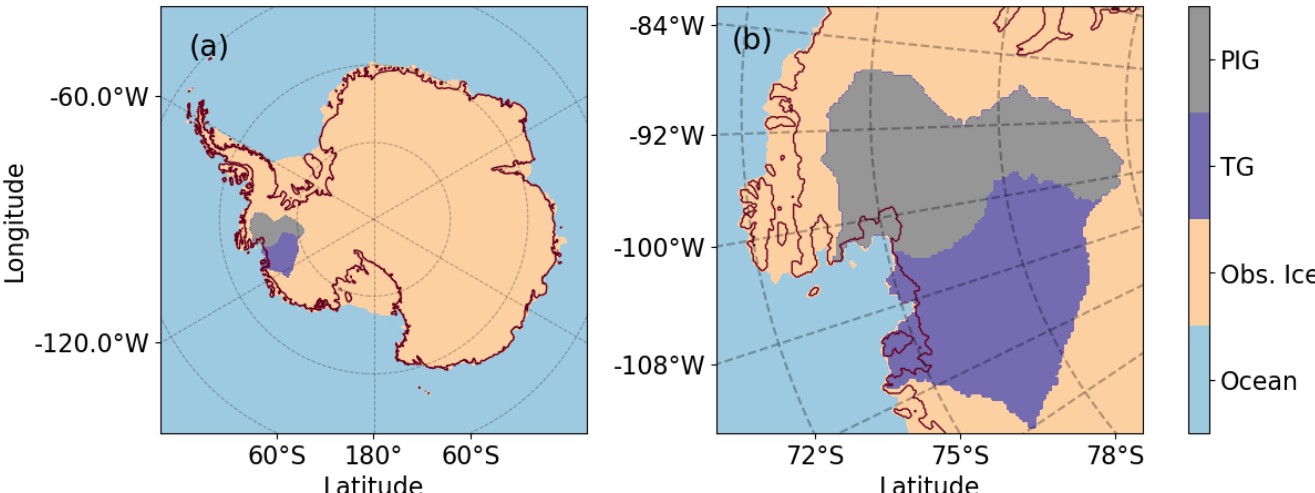

**Figure 2. Schematic overview of the modelled area**. The Antarctic Ice Sheet (a), with orange denoting the region where ice is observed in the dataset of Morlighem et al. (2020). The observed grounding line (following Morlighem et al. (2020) and applying hydrostatic

equilibrium) is shown by a thin red line. The TG basin is shown in purple and the PIG basin in grey, following Zwally et al. (2015). Basin-integrated calculations are applied over these two areas. (b) a close-up of the Amundsen Sea Embayment.

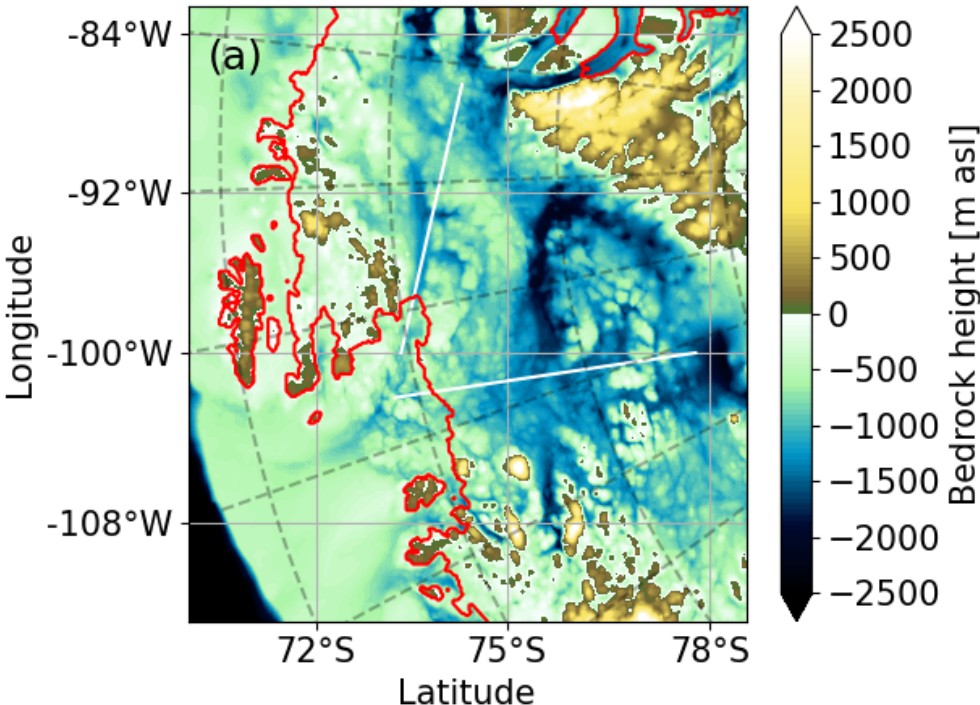

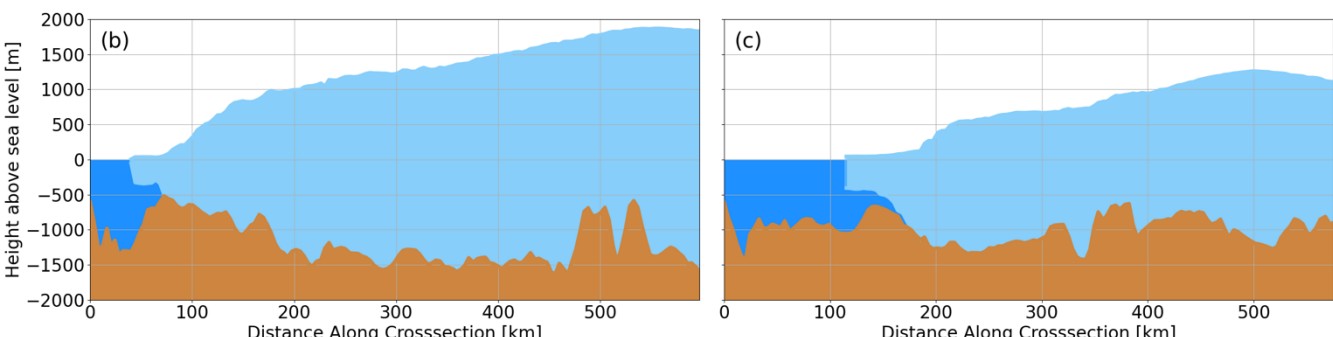

Figure 3. Regional setting of TG and PIG. (a) bedrock profile of the Amundsen Sea Embayment with the observed grounding line position, using ice thickness and bedrock height observations from Morlighem et al. (2020) in red. White lines indicate the locations of the cross sections shown below. (Bottom row) cross sections with the ice sheet shown in light blue, the ocean in dark blue, and bedrock in brown. TG in (b) and PIG in (c).

## 4 Results

In this section, we first show and discuss the modelled present-day ice sheet using the two initialization methods (Sect. 2.4). We highlight key differences and discuss implications of choices made during the initialization. Then we present the unforced future simulations, discussing their different responses to changes in basal friction and showing how these differences are related to the ice sheet geometry and buttressed ice shelves.

### 4.1 Initial condition evaluation

As a starting point for our forward experiments, we use the two spin-up types described in the previous sections. We evaluate the initial states here.

#### 4.1.1 Default initialization (DI)

Figure 4 shows the initial ice sheet state after the default initialization. The overall thickness bias is low. The regional thickness bias of the East Antarctic Ice Sheet (EAIS) relates to the small observed thickening in central EAIS (Smith et al., 2020), which equals a mass flux similar in magnitude to the local surface mass balance. The RMSE between modelled and observed ice thickness and modelled and observed ice surface velocity are respectively 21.10 m and 135.81 m/yr.

The modelled grounding line position (Fig. 4a) matches the observed position well, with a modest average error of 1.4 km. Surface ice velocities generally agree with observations except for glaciers on the Siple coast, which flow slightly too fast, and the seaward sides of the Filchner-Ronne and Amery ice shelves, where the flow is too slow. Assuming that the observed imposed $dH/dt$ is correct, this implies that the ice flux along flowlines in these locations decreases too quickly in CISM. Hence, to retrieve the observed geometry during the inversion, basal melt is decreased. The inverted $C_c$ (Fig. 4c) is generally high in the interior or under slow-moving areas of the ice sheet, and low under outlet glaciers. The inverted ocean temperature perturbations (Fig. 4d) under the larger ice shelves (Filchner-Ronne, Ross and Amery) are generally close to zero with the exception of some positive corrections under the PIG, TG, and Crosson shelves in the ASE region.

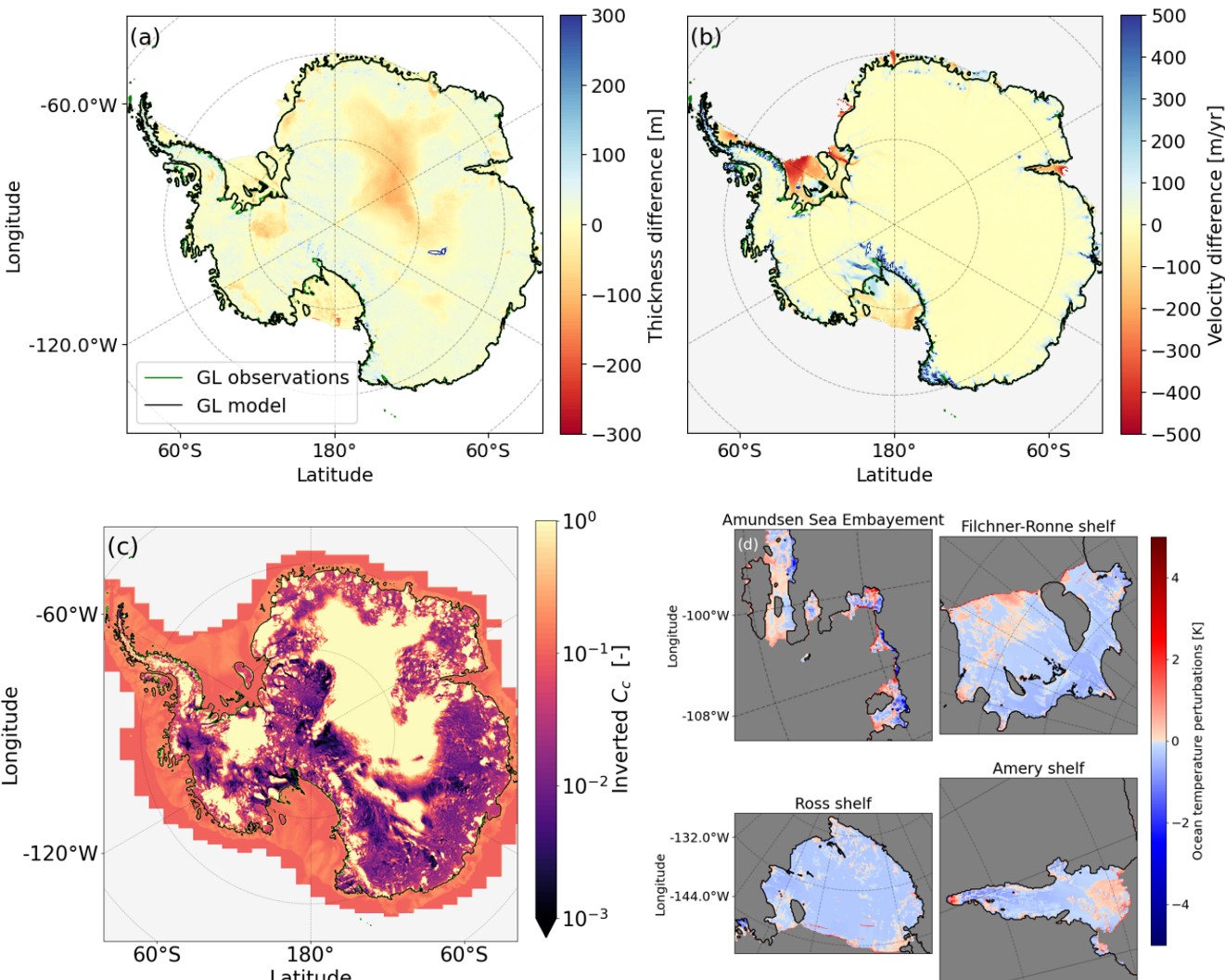

**Figure 4. Modelled Antarctic Ice Sheet initialized state with the default inversion (DI).** (a) thickness difference with respect to observations (Morlighem et al., 2020). The modelled grounding line is shown in black, and the observed grounding line in green (only visible where it does not overlap with the modelled one). (b) ice surface velocity difference with respect to the observations (Rignot et al., 2011). Positive values indicate regions where CISM overestimates the ice velocities. (c) the inverted $C_c$ from Eq 1.1 using Eq 1.12. and (d) the inverted ocean temperature perturbation under the main shelves.

### 4.1.2 Flow Enhancement Factor Initialization (FEFI)

Figure 5 shows the ice sheet state after the FEFI 10 kyr initialization. Optimizing for both ice thickness and ice velocity creates a trade-off leading to a slightly increased thickness error and generally a decreased velocity error (RMSE thickness and velocity: 45.99 m, 101 m/yr ). The thickness bias in the interior of the EAIS has grown in area and magnitude, showing that either the provided SMB is too low, or the observed and imposed thickening is overestimated. While the thickness bias in the

interior of the EAIS has persisted, the bias for the outlet glaciers has increased in magnitude. This suggests that either the
prescribed surface mass balance (SMB) and/or dynamic mass loss (through the inclusion of dH/dt) is too low. This effect is
evident in the inverted ($C_c$) field: in the FEFI simulation, outlet glaciers in Dronning Maud Land all reach a value of 1. At this
point, the friction inversion can no longer counteract ice loss, and the enhanced flow speeds increase the ice flux, thinning the
ice and further amplifying the thickness bias.

There is no large differences in the inverted $C_c$ (Fig. 5c) compared to Fig 4c, and the inverted ocean temperature
perturbations show the same pattern as in Fig. 4d. Ocean temperature perturbations are generally larger near the calving front
and lower in the interior of the shelves, especially for the Filchner-Ronne and Ross shelves. Ice velocities are generally
greater in the shelves compared to the default inversion, better matching the observed velocities. This increases the ice flux
through the ice shelves, lowers the need for basal melting in the shelf interior, and increases the ice flux at the calving front.
The thickness misfit over the ice shelves increases slightly and is predominantly negative, indicating that the modeled
shelves are generally too thin. This is due to the flow enhancement factor yielding increasing ice velocities, which in turn
raises the flux toward the calving front. Ocean temperature inversion can only partly compensate for this thinning by
reducing basal melt rates because once the inverted ocean temperature perturbation equals the negative of the thermal forcing
and their sum becomes zero, basal melt cannot decrease further as Eq. 1.13 does not allow for accretion. If the flow
enhancement factor continues to accelerate ice flow beyond this point, the shelves will continue to thin.

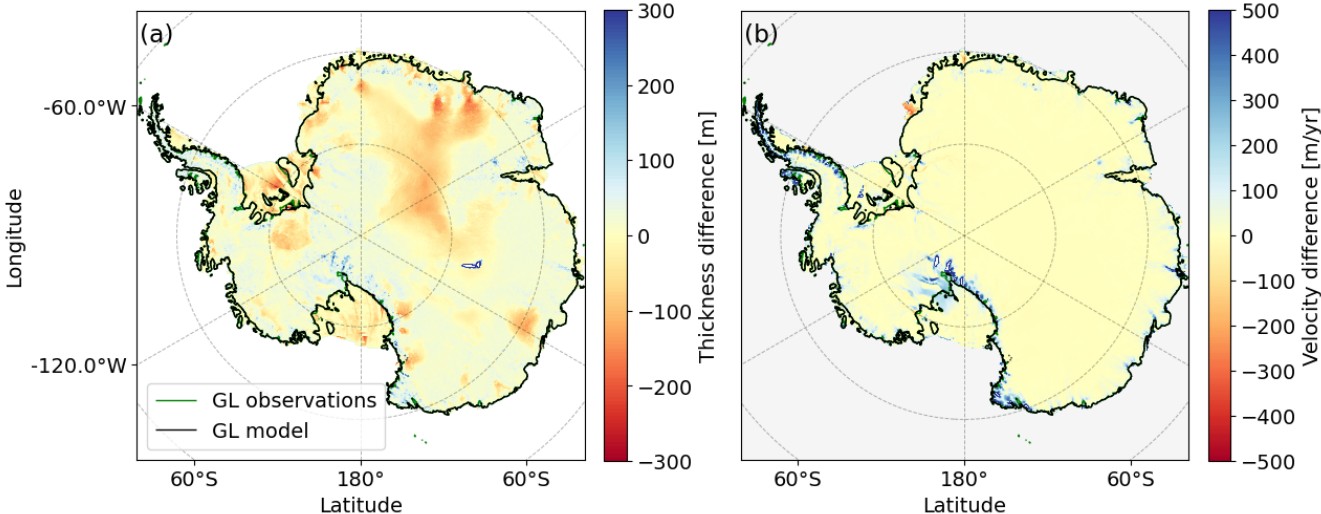

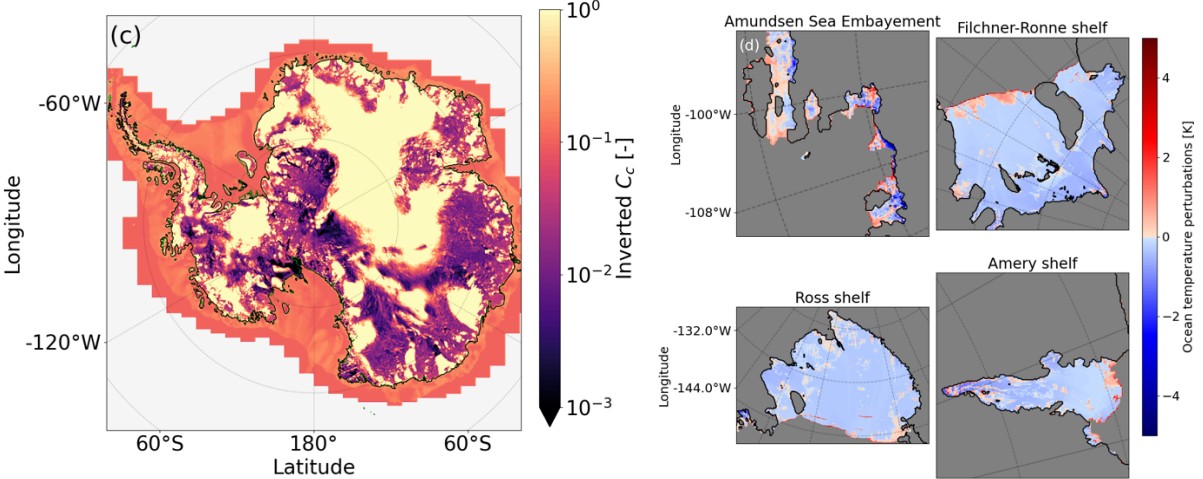

**Figure 5. As in Figure 4 but for the flow enhancement factor inversion FEFI.**

Initializing the model with the DI approach results in a dipole misfit in surface ice velocities at the TG grounding line: the Eastern Thwaites glacier flows too slowly, while the Western Thwaites glacier is too fast (Figure S10). A similar dipole pattern emerges near the PIG grounding line, where the model overestimates velocities along the shear margins but underestimates flow along the main trunk. These discrepancies are likely due to the absence of damage representation in the shear zones, an effect that, if included, would increase the velocity gradient in the shear zone, allowing for an even more sharply defined ice stream bounded by near-stagnant ice (Lhermitte et al., 2020; Izeboud and Lhermitte, 2023).

It is important to note that the integrated grounding line fluxes shown in Figure S11 for the DI and S12 for FEFI are close to observational estimates. This suggests that the slower main flow of PIG and the faster-flowing shear margins approximately balance out, resulting in a total ice flux that matches observations, an outcome that also holds for TG. Applying FEFI significantly reduces the velocity misfits, though the dipole patterns in ice surface velocity misfit persist because the flow enhancement factor inversion is constrained to a minimum (0.1) and maximum (10) value, which is reached at a few spots in these regions (see Fig S14 and Fig S15). Nevertheless, the integrated ice fluxes through the main flow lines of both glaciers remain close to observed values, even slightly improved compared to the DI initialization. While we acknowledge the surface velocity errors and their potential influence on unforced simulations, the agreement in ice fluxes supports the use of both initialization methods, DI and FEFI, for future unforced model runs.

When initializing an ice sheet model using observed ice thickness and surface velocities, it is essential to incorporate present-day mass change rates, particularly in regions experiencing the highest thinning. Omitting these rates, thus assuming dH/dt = 0, and still tuning the model to match observed surface velocities leads to compensating behavior. The dynamic mass loss contributes considerably to the ice fluxes of TG and PIG, and neglecting this flux contribution creates a profound mismatch

between the observed thickness and surface ice velocity, as the product of both equals the ice flux. The model will increase surface velocities by lowering the ice viscosity, and to prevent ice thinning, it will raise basal friction. In essence, if the momentum balance equations are accurate and the observational data reliable, then prescribing both the observed ice thickness and surface velocities should naturally reproduce the observed mass change pattern. Ignoring the latter and assuming zero mass change forces the model to compensate by introducing systematic errors.

Figure S14 shows the inverted flow enhancement factor and the velocity error change, basal friction change and flow regime change between FEFI and DI simulations for the Amundsen Sea region. First, the flow enhancement factor $E$ in Eq 1.7 is not spatially constant anymore but shows a chaotic pattern in the main flowlines of PIG and TG. PIG and West TG generally weaken (higher $E$), while East TG stiffens. The velocity differences are due to a local inversion (cell-by-cell $E$ inversion) and a highly non-local influenced variable (ice surface velocities). The basal friction shows a similar speckled pattern. The flow regime is quantified by dividing the ice basal velocity magnitude by the ice surface velocity magnitude. A factor below 1 shows deformation-dominated (SIA) flow, and a factor of 1 shows no vertical velocity shear and therefore sliding-dominated (SSA) flow. Remarkably enough, in large areas close to the grounding line, the flow regime becomes more deformation-dominated. Where $E$ increases, the ice becomes less viscous, and the flow regime becomes more favorable to deformation.

This is particularly striking at the Western TG grounding line, where at present the regionally highest ice surface velocities are observed, which are unlikely to be solely by deformation. Sliding is expected to be the dominant regime of a fast-flowing (Antarctic) outlet glacier from standard ice flow theory, and the SSA is widely used as the appropriate stress approximation to model these regions (e.g. Bueler and Brown (2009); Brondex et al. (2019); Gudmundsson et al. (2023); Morlighem et al. (2024)). However, Mccormack et al. (2022) modelled the ice flow regime more extensively and more physically than we do, and found a heterogenous pattern of sliding and deformation close to the TG grounding line depending on the flow law used. Therefore, a mix of sliding and deformation cannot be excluded entirely (see Fig 2 in Mccormack et al. (2022)).

Because our model uses Glen's flow law (which was developed for isotropic ice flow and secondary creep) it cannot capture e.g. tertiary creep and ice damage accurately (Glen, 1952; Budd et al., 2013; Graham et al., 2018). By inverting viscosity, we are effectively compensating for this missing process, so the resulting flow-enhancement factor and inferred flow regime reflect model deficiencies rather than intrinsic ice properties. This is particularly true in regions with fast-flowing ice, because our FEFI initialization is only allowed to change the flow enhancement factor in areas where the modelled ice surface velocity errors exceed 25 m yr$^{-1}$. In addition, FEFI assigns ice properties to fixed grid cells instead of advecting them along flowlines, even though impurities, damage, and fabric anisotropy are fundamentally Lagrangian properties of the material. If the ice is damaged, it will remain so downstream of where the damage was initiated. The FEFI inversion can therefore generate physically questionable enhancement factors that mask upstream errors in the flow regime. For these reasons, we doubt that deformation dominates at the TG grounding line, since the current inversion cannot yet reproduce physically consistent ice

properties. Observations could provide clarity on the flow regime of the TG grounding line. In particular, measurements along the vertical profile of the horizontal velocity in critical regions will help to distinguish which flow regime dominates.

We would therefore reiterate that we do not see the FEFI results as physically plausible inverted properties but rather as the result of model choices made during the initialization procedure. We use FEFI to generate a similar initialized AIS state as for

the DI, that responds differently to modellers choices made regarding the basal friction parameterization. Those differences, and the general results of our continuation experiments, are presented in the next paragraphs.

## 4.2 Modelled unforced evolution of WAIS

**Figure 7. Sea level contributions from the ASE for four different sliding laws** with basal friction and ocean temperature perturbation inversion (solid lines) and including the flow enhancement factor inversion (dashed lines). The ice volume above floatation [VAF] percentage

change is the loss of ice that can contribute to sea level change (i.e., that is not already floating or present below sea level), relative to the beginning of the simulation, for the basins containing PIG and TG, respectively basin 22 and 21 in Zwally et al. (2015) . The solid black line is the interpolated present-day trend (Smith et al., 2020).

Figure 7 shows the global mean sea level contribution of eight simulations initialized with the mass change rates from Smith et al. (2020). In all simulations, both big glaciers in the ASE eventually collapse and most of the ice volume above floatation (VAF) is released to the ocean. These results are comparable to the results presented in Van Den Akker et al. (2024). In general, the simulations starting from FEFI simulate a longer period of linear mass loss before the start of a steep decline in VAF. For the DI-starting simulations, two stages can be identified. These are i) linear VAF loss similar to the present-day rate and ii) a simultaneous collapse of PIG and TG starting around year 400. For simulations starting from the FEFI, three stages can be identified, in contrast with the default initialization (solid). These are i) a linear decline in VAF similar to the present-day rate for the first 600 years, ii) PIG collapse for 300 (Schoof and Zoet-Iverson) or 600 (power law and pseudoplastic) years, and iii) TG collapse for approximately 200 years. The maximum rate of sea level rise during the third (TG collapse) phase differs marginally among the eight simulations ($4 \pm 0.7$ mm GMSL per year).

The simulations starting from FEFI exhibit behavior in line with the results of Brondex et al. (2017); Brondex et al. (2019), and Sun et al. (2020): Regularized Coulomb sliding (Schoof and Zoet-Iverson) yields earlier and faster collapse than simulations with pseudoplastic and power law sliding. This is in stark contrast to the DI simulations, in which the rate of glacier collapse is much less affected by the choice of the basal friction parameterization. The DI results show the relative non-sensitivity of ice sheet modelling to the specific basal friction law, agreeing with the results of Barnes and Gudmundsson (2022) and Joughin et al. (2024) .

### 4.2.1 DI: Collapse mechanics and characteristics

We start discussing the results from the DI experiments (solid lines in Fig. 7), where the collapse rate shows little sensitivity to the choice of basal friction law. Figure 8 presents elevation profiles of PIG (top row) and TG (bottom row) along the cross-sections indicated in Fig. 3. Before the collapse (year 250, first column), differences in ice sheet geometry and grounding line position across the four sliding laws are minimal for both glaciers. The final states after collapse (year 750, last column) also appear similar. However, at the midpoint of the collapse (year 500, middle column), the glacier shapes diverge for TG (panel b). For example, in the simulation using power law sliding, the grounding line has retreated roughly 75 km further than in the Zoet-Iverson sliding case. Notably, the ice upstream of the grounding line is thinner for the Zoet-Iverson sliding case compared to the power law, but the latter shows more grounding line retreat. The Zoet-Iverson sliding case loses more mass further inland, and the powerlaw sliding case loses mass close to the grounding line and has a thinner ice shelf. From Fig 7 we conclude that the integrated mass loss and SLR from the DI sliding law simulations is very similar, but from Fig 8 we see different geometries during the TG collapse.

This difference in geometry can also be seen in Fig. 9, which shows spatial patterns of thickness change and grounding line position. The grounding line of the power law simulation (top row) is retreated further inland compared to the Zoet-Iverson simulation (bottom row) during the collapse at year 500 (middle column). After 750 years, when the collapse has happened
and the mass loss slows down, both simulations show a similar geometry again. These experiments show that the Zoet-Iverson sliding law, with lower friction far upstream of the grounding line, leads to more ice being advected from inland towards the grounding line compared to the power law. As a result, the Zoet-Iverson simulation shows less thinning and retreat near the grounding line, but more inland thinning, compared to the power law simulation. This means that the Zoet-Iverson sliding law results in a smaller but thicker ice shelf in front of the collapsing TG, with a higher buttressing potential and a greater chance
to stick to pinning points.

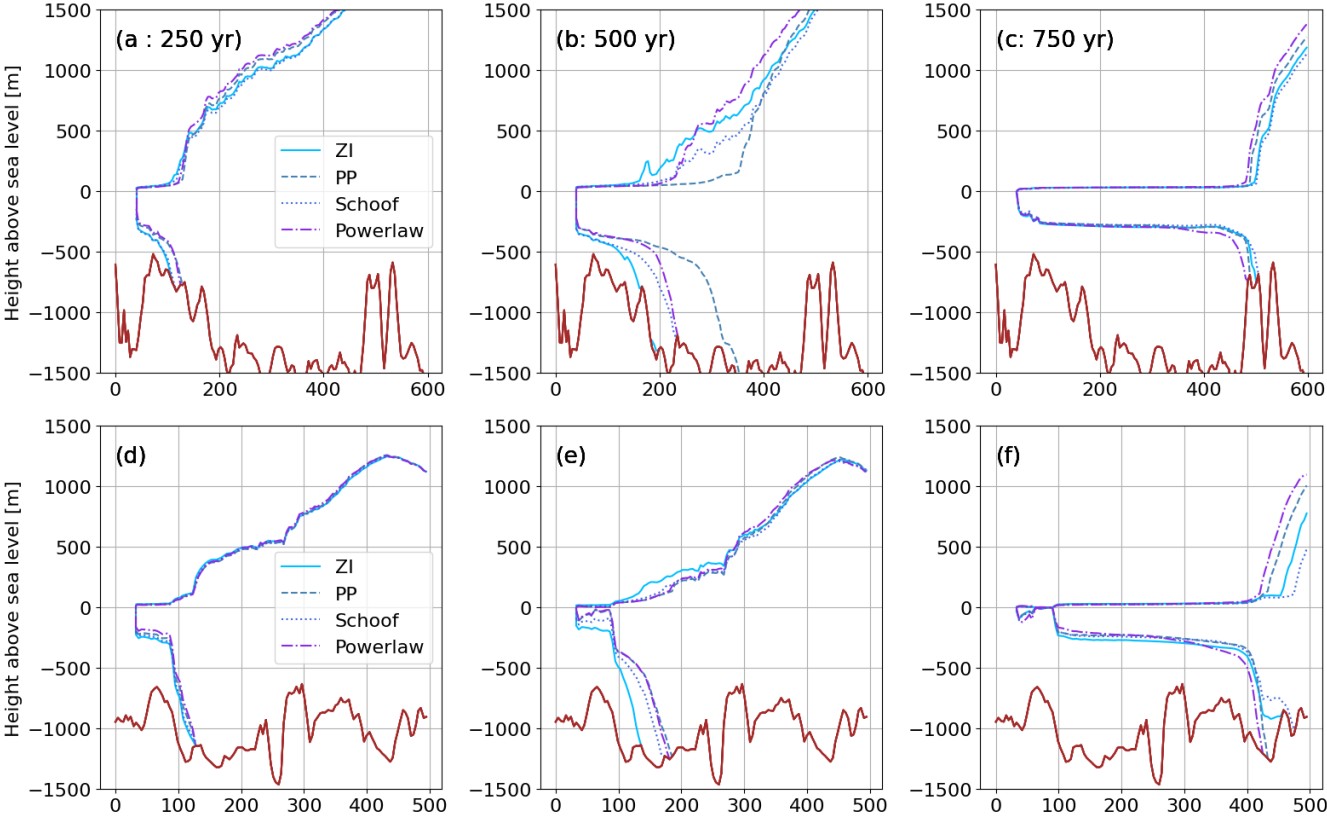

**Figure 8. Default Inversion (DI) retreat patterns for** (TOP ROW) Thwaites Glacier along the cross-section shown in Fig 3 at simulation
515    years 250 (left column), 500 (middle column) and 750 (right column) for Zoet-Iverson (solid), pseudoplastic (dashed), power law (dashdot) and Schoof (dotted). Bottom row d-f, as a-c but for Pine Island Glacier. Bedrock height asl is shown in brown.

The similarity between the simulations with Schoof and Zoet-Iverson sliding, and between those with power law and pseudoplastic sliding, can be explained by their similar functional relations between basal velocity and friction, shown in Figure 1. In the rest of this section, we focus on one member from each pair: the Zoet-Iverson law and the power law.

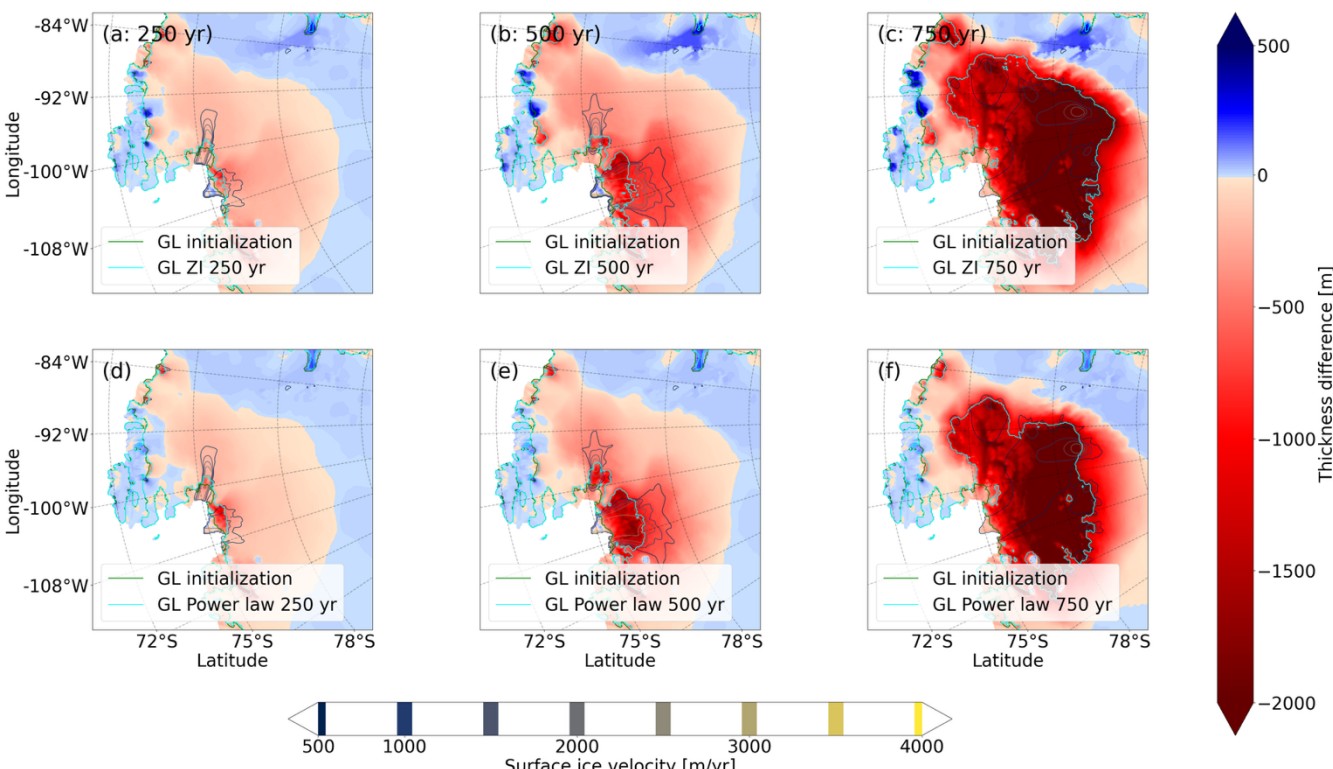

**Figure 9**. **Default inversion (DI) spatial retreat patterns in the ASE region.** Top row: Zoet-Iverson thickness change since the initialization at year 250 (a), 500 (b) and 750 (c), with grounding line position (green: initialized, cyan: at the timestamp of the figure) and surface velocity contours (every 500 m yr$^{-1}$. Bottom row: same for the powerlaw.

To explain the difference in collapse mechanism but a similar VAF evolution, we analyze the buttressing of TG and PIG during the collapse. We applied both buttressing quantifications described in section 2.2 to the 250- and 500-year Zoet-Iverson and power law simulation, as shown in Figure 10. Both simulations show a less buttressed Western Thwaites Ice Stream and more buttressed Eastern Thwaites Ice Stream. Moving closer to the calving front decreases the buttressing number. In general, the buttressing close to the grounding line is stronger for the Zoet-Iverson sliding law, according to this method. Little difference can be seen in the confined Pine Island Glacier in both simulations.

The right side of Figure 10 shows acceleration factor during the shelf-removal experiments at years 250 and 500, comparing Zoet-Iverson sliding (left column) and power law sliding (right column). Following the removal of the ice shelves, grounded ice in the Zoet-Iverson simulation accelerates rapidly, especially at the TG grounding line and further inland, much more so than in the power law case. Two primary mechanisms can slow down a retreating marine-terminating glacier like TG: buttressing and basal friction. At year 500, buttressing at the TG grounding line is stronger in the Zoet-Iverson simulation than in the power law case, as indicated by the higher buttressing numbers. Despite this, the acceleration response to shelf loss (Fig. 10c,d,g,h) is greater in the Zoet-Iverson case. This is because, in the power law case, basal friction increases with velocity (as shown in Fig. 1), limiting the glacier's speed-up and upstream propagation of the acceleration. In contrast, the Zoet-Iverson simulation exhibits less frictional resistance and thus stronger acceleration. This is further examined in Figure S16 and S17, where we show the ice velocity increases due to ice shelf removal when the Zoet-Iverson geometries are tested using power law sliding, and the power law states are tested using Zoet-Iverson. To change sliding law while retaining the exact same ice sheet state and velocities prior ice shelf removal, we used again the procedure described in the Supplementary Materials. Now, the power law states with Zoet-Iverson basal friction have the largest velocity increases (Fig S17), even larger than these for the Zoet-Iverson state using Zoet-Iverson (Fig. 10e,g). Conversely, the Zoet-Iverson states with a power law leads to the lowest velocity increases (Fig. S16), which are also lower than these for the power law state using the power law (Fig. 10f,h). We conclude that, during the TG collapse, buttressing is the primary braking mechanism in the Zoet-Iverson case, whereas increased basal friction dominates in the power law case. Interestingly, due to the specific bed geometry of TG and the conditions in the unforced simulation, both scenarios produce a similar contribution to global mean sea level rise—though driven by different mechanisms.

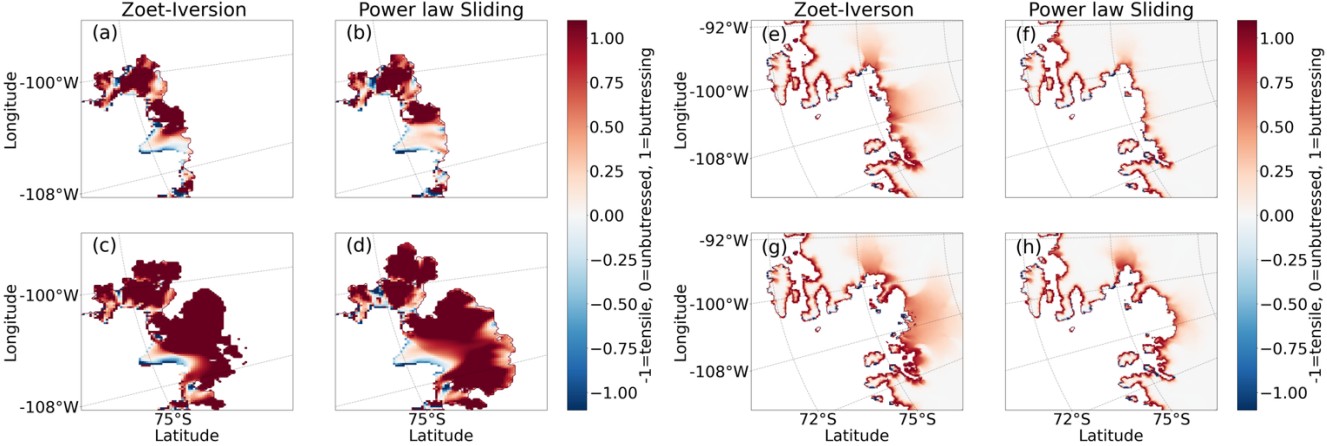

Fig 10. **Buttressing number and acceleration number for the DI simulations (**left four panels) Buttressing number calculated over the floating ice shelves at 250 years (top row) and 500 years (bottom row), for the Zoet-Iverson sliding law (left column) and the power law (right column). (Right four columns) acceleration number after removing the ice shelves at 250 years (top row) and 500 years (bottom row),

for the Zoet-Iverson sliding law (left column) and the power law (right column). Note the different zooms of the four panels on the left and the four panels on the right. This is done to preserve detail in the left four panels.

With regularized Coulomb sliding in our DI continuation simulations during the TG collapse, the dominant resistive force is buttressing, whereas with power law sliding, the dominant resistive force is basal friction. There is a compensation effect visible in our continuation experiments because the ice shelf in all runs is allowed to persist (no calving front retreat, no forcing applied other than the present-day calibrated ocean temperatures). If the shelf were significantly weaker because of either

calving or ocean warming, we would expect the Zoet-Iverson law to yield faster collapse, since the buttressing would not be present to compensate the lower friction.

### 4.2.2 FEFI: Collapse mechanics and characteristics

In contrast to the default experiments, the simulations starting from the FEFI show a strong sensitivity to the choice of basal friction law in terms of integrated ice mass loss and GSML rise contribution. Also, the collapse of the ASE occurs later,

beginning around year 800 in the Zoet-Iverson case and around year 1200 in the power law case, and it follows a different pattern. Instead of transitioning directly from present-day-like mass loss rates (~0.3 mm/yr) to full collapse rates (~3 mm/yr), there is an intermediate phase with mass loss rates of approximately 1–2 mm/yr. This transitional phase is driven primarily by the collapse of PIG, which occurs independently of TG in the FEFI simulations.

Figure 11 shows a typical snapshot from the collapse phase (the first inflection points in Fig. 7) in both initializations. In the simulations starting from the DI, TG collapses first. In contrast, PIG collapses first in the FEFI simulations, while TG is temporarily stabilized on a bedrock ridge about 40-50 km upstream of the present-day grounding line (white line in Fig 11). TG reaches this grounding line quickly, before PIG's grounding line starts to recede, but then stabilizes for 800 – 1000 years on a local high in the bedrock (white line in Fig 11.). Once PIG has retreated sufficiently and starts to draw ice from the

Thwaites basin, TG starts to collapse. Thus, unlike the DI simulations, PIG initiates ASE deglaciation when using FEFI.

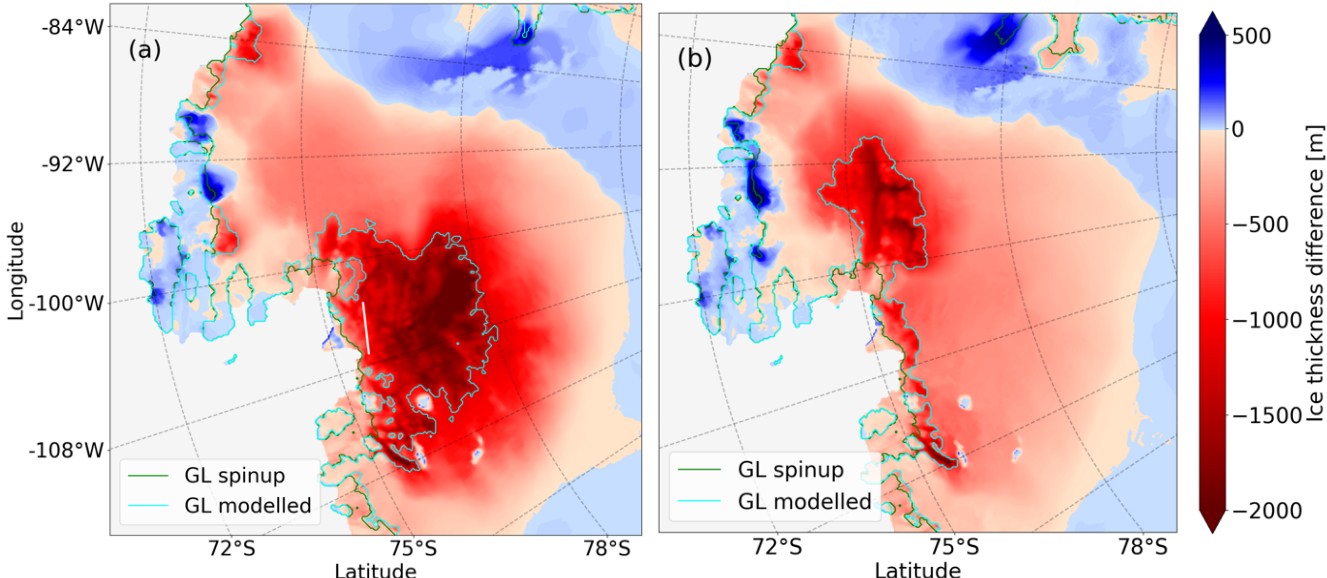

**Fig 11. Start of Amundsen Sea ice collapse in simulations with the DI (left) and FEFI (right).** Colors indicate the ice thickness difference with respect to the initialization, and the modelled grounding line is shown for the initialization (green) and continuation (cyan). The time snapshots of the left and right panels are respectively for years 575 and 775. Results shown here are from the Zoet-Iverson sliding simulations, but results are similar for the power law simulation. The white line in figure (a) highlights the ridge identified in Van Den Akker et al. (2025). As soon as the ice ungrounds from this ridge, a collapse of TG is imminent.

PIG often collapses more slowly than TG in our simulations. This is shown in Fig. 12. A typical collapse phase of the PIG in these simulations lasts about 300 years when using a Zoet-Iverson sliding law, and up to 800 years when applying power law friction. During this time, the ice sheet loses about 50 centimeters in GMSL equivalent.  Since we do not apply surface melting, all losses of grounded ice happen through advection over the grounding lines. A large ice flux over the grounding line initially thickens and strengthens the ice shelf. For TG using the DI simulations, this cause braking effect: a thicker, stronger and more buttressed ice shelf slows down the upstream flow and lowers the ice flux through the grounding line. For PIG, the increased grounding line flux apparently does not lead to a sufficient increase in buttressing to slow down the collapse.

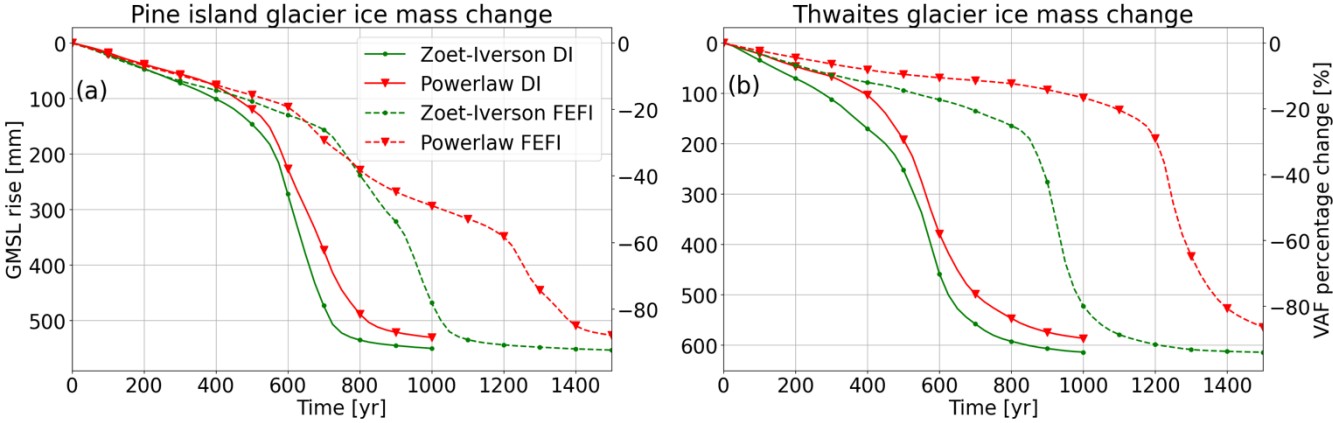

**Figure 12.** Mass loss shown separately for PIG (left, a) and TG (right, b), for both initializations (solid is DI and dashed is FEFI) and for two sliding laws.

**Figure 13. Ice shelf removal experiments and buttressing quantification during the FEFI continuation simulations**. The top row shows buttressing values for the Zoet-Iverson (left) and power law (right) cases, while the bottom row displays ice velocity accelerations resulting from the shelf-removal experiments. Note that the grounding line positions differ slightly between the top and bottom rows, despite only a single model timestep (2 months) separating them. This discrepancy is due to the rapid grounding line retreat occurring during the collapse.

Figure 13 shows results for the shelf-removal experiments and the buttressing number, repeated with the FEFI initialization. We remove the shelves just before the accelerated collapse begins, at 775 years into the simulation. We can see that an effect is visible: close to the grounding lines, the simulations with the Zoet-Iverson sliding law have a larger speed-up. In the power law case, removing the shelf matters less. However, the difference in acceleration as a reaction to the removal of all floating ice is smaller for the FEFI initialization, compared to the DI initialization, when comparing the size and inland extent of the acceleration number in the main flow line of TG with the effect in Figure 13 around the GL of PIG. This is reflected in the similar buttressing numbers for the two sliding laws states. Hence, the ice shelves formed during the PIG collapse in the FEFI simulations are weaker than the shelves formed during TG collapses in the DI continuations, and they cannot brake the increased ice velocities of the Zoet-Iverson simulations as they did in the DI simulations with TG collapse. At this stage of the FEFI simulations, TG has retreated into a confined embayment with pinning points, with a slightly stronger shelf in the Zoet-Iverson sliding simulation.

In summary, the FEFI simulations show more sensitivity to the choice of basal sliding law than the DI simulations. This is related to the existence of a strong buttressing TG shelf at the start of the collapse in the DI simulations, whereas there is no strong buttressing PIG shelf at the start of the collapse in the FEFI simulations. In our simulations, the initialization determines whether the collapse starts with TG or PIG, but also other factors, like differences in future ocean warming, can influence which basin fails first. Therefore, the specific ice sheet evolution during the collapse determines the sensitivity to the choice of basal friction law.

**5 Discussion**

Our study is consistent with studies arguing that different basal friction parameterizations cause significantly different response (Brondex et al., 2017; Sun et al., 2020; Brondex et al., 2019), and also with studies that claim the opposite (Barnes and Gudmundsson, 2022; Wernecke et al., 2022), although the latter two studies focus on shorter timescales (~100 years) and feature much less grounding line retreat than our study (e.g., no WAIS collapse). We argue that sensitivity to the sliding law depends on the geometric evolution during the retreat phase, e.g. on whether newly formed ice shelves can survive and provide considerable buttressing. In our cases, the geometric evolution is sensitive to modellers choices made during

initialization, even though these two initial states are very similar. This study was inspired by Berends et al. (2023), who showed that obtaining a similar initial state does not necessarily lead to the same forced retreat in idealized experiments.

The connection between buttressing and basal friction during TG collapse hinges on the survival of the ice shelf that forms during grounding-line retreat. This is in turn determined by ice flux over the GL and available pinning points, but also on the basal melt and calving rates. With respect to calving, we apply a no-advance calving front at the present-day position. Theoretically, the calving front can move inland, but we use a conservative limit of 1 m ice thickness before ice is allowed to be removed. In practice, this rarely happens. Using a physically-based calving law would likely increase calving rates as the

ice thins and the grounding line retreats, and might influence the compensating feedback demonstrated in this study.

With respect to basal melt rates, we apply the ISMIP6 basal melt parameterization (Seroussi et al., 2020) with thermal forcing data from Jourdain et al. (2020). Although this approach provides basal melt fluxes in agreement with observations and other model studies (see Van Den Akker et al. (2025)), other approaches — such as including a coupled cavity-resolving (regional)

ocean model or a sub-model capturing cavity flow like PICO (Reese et al., 2018b) — could result in basal melt rates that are more physically based and lead to different results. The ISMIP6 parameterization lacks freshwater feedbacks such as the reduced formation of Antarctic Bottom Water (Williams et al., 2016) and cooling of the sea surface (Bintanja et al., 2015). Increases or decreases in future basal melt rate will moderate the effective buttressing of the newly formed shelves. Resolving basal melt rates with a model for ocean circulation in cavities could be an interesting topic for future research.

This study accounts for subglacial hydrology only in a simplified way. We parameterize the effective pressure N according to Leguy et al. (2014), where we assume that N is reduced near grounding lines because of a connection between the subglacial hydrology network and the ocean. While this captures some aspects of including a subglacial hydrological system (e.g. lowering basal friction in areas close to grounding lines), it does not simulate a complex hydrological network as was

655 done, for example, by Kazmierczak et al. (2024) or Bradley and Hewitt (2024). Including a more complex hydrological network and coupling it via the effective pressure to the basal friction would likely alter our results because hydrological processes are now incorporated in the basal friction inversion. Taking those out of the inversion will likely change the inverted fields considerably, and therefore also our projections.

When future ocean warming is applied, the resulting ice shelves in the ASE are expected to be much smaller. As a result, the

660 buttressing effect that moderates retreat in the Zoet-Iverson simulation would be reduced, likely leading to greater projected global mean sea level (GMSL) rise compared to the power law case. Therefore, the results presented here are specific to the CISM model, the initialization techniques used to reproduce present-day mass loss, and the absence of any future forcing. Follow-up studies are needed to evaluate whether our conclusions hold under different scenarios, such as schematic ocean warming or through transient calibrations designed to reproduce historical mass loss trends (e.g. Goldberg et al. (2015)

The flow enhancement factor tuning is directly influenced by the choice of the momentum balance and the use of simple generalized flow law (Eq 1.18). Rathmann and Lilien (2022) show that the tuned flow enhancement factor when using Eq 1.18 compensates for more complex ice fabric properties only when the basal friction coefficient is known. This is not the case in our study, so our inverted flow enhancement factor has no physical meaning but is just a bias correction term and was necessary to obtain a new initialization to compare our DI results with. However, it reduces the velocity error misfits considerably. Future work could focus on parameterizing shear induced anisotropy and damage, repeating the experiments done by Rathmann and Lilien (2022), implementing a more complex anisotropic flow law proposed for example by Gillet-Chaulet et al. (2005), and/or simulate the full Stokes momentum balance.

Our inverted flow enhancement factor results generally align with the heterogeneous pattern of deformational flow reported by Mccormack et al. (2022) with the notable exception of a localized patch of deformation-driven flow at the TG grounding line in our simulations. Barnes et al. (2021) examined the transferability of inverted parameters across three ice sheet models and found substantial variation in the inverted rate factor among them. Although a direct comparison with our inverted flow enhancement factor is difficult, since the rate factor also depends on temperature (see Eq. 1.16), the inverted rate factors in the Úa model (see Fig. 4 in Barnes et al. (2021)) vary by up to two orders of magnitude. This is consistent with the heterogeneous patterns we observe in our own results. Comparison with more recent model studies employing some kind of viscosity inversion (Hill et al., 2021; Dawson et al., 2022; Bradley et al., 2025), was not possible since those studies listed do not present an inverted viscosity and/or flow enhancement factor field

The FEFI initialization reduces the misfit between modelled and observed ice surface velocities in the ASE compared to an initialization where only ice thickness is used as target variable, which may partly explain why experiments starting from the FEFI state show a delayed retreat compared to those initialized with DI. This delayed retreat allows basal friction to play a greater role in controlling the rate of grounding line retreat. The disadvantage is the addition of another free, unconstrained, parameter. This makes the system of equations more underdetermined. It would be useful to have observations (like ice velocity depth profiles) on where the ice sheet flow is deformation or sliding dominated, especially in key regions like at the present-day Thwaites grounding line or the pinning point 40 km upstream. Observations could include strain meters in boreholes or surveys of the ice basal velocities in key regions.

## 6 Conclusion

In this study, we conduct Antarctic Ice Sheet simulations initialized to be consistent with present-day mass loss rates, in which Thwaites Glacier and Pine Island Glacier collapse. We use two initializations; one initialization that solely inverts for basal friction coefficients based on the mismatch between modelled and observed ice thickness, while the other also inverts for a flow enhancement factor based on the mismatch between modelled and observed surface velocities. These two intializations

lead to two distinctive sets of future ice sheet evolutions. In the former inversion, Thwaites glacier collapses first and exhibits a connection between basal friction and buttressing: increased ice velocities and grounding line fluxes can increase the buttressing capacity of the ice shelf downstream. This makes the future projections from this inversion in this study insensitive

to the specific basal friction law used. In the latter inversion, Pine Island Glacier collapses first, which does not exhibit a connection between basal friction and buttressing: here employing power law friction slows down the eventual collapse. As a result, the sensitivity of our modelled Antarctic Ice Sheet to the choice of basal friction parameterization is determined by the order of collapse, which in turn is determined by the initialization.

The results presented in this study illustrate why CISM can generate evolutions with either clear or weak sensitivity to the choice of basal sliding law, as ice shelf buttressing can potentially, but surely not necessarily, provide a negative feedback on the grounding line fluxes. Carrying out these types of experiments with other ice sheet models will enhance our understanding of why some simulations (Brondex et al., 2017; Sun et al., 2020; Brondex et al., 2019), are more sensitive to changes in basal friction laws than others (Barnes and Gudmundsson, 2022; Wernecke et al., 2022), and possibly lead to similar conclusions.

A potential way to address this issue is through standardized tests (e.g. MISMIP+ for different sliding laws) after major changes have been made to the initialization procedure of the ice sheet model. Also, projections could start from an ensemble of many different initializations, all done with different model choices (e.g., inverting for the flow enhancement factor or not, as was done in this study). Explorations with more realistic treatments of calving and ocean thermal forcing could also be illuminating. Finally, new (depth) observations on the relative strength of sliding- versus deformation-dominated flow would decrease the

degrees of freedom now present in the initialization procedures of ice sheet models.

**Code availability**

CISM is an open-source code developed on the Earth System Community Model Portal (ESCOMB) Git repository available

at https://github.com/ESCOMP/CISM. The specific version used to run these experiments is tagged under https://github.com/ESCOMP/CISM/releases/tag/CISM_basalfriction_buttressing_version .

**Data availability**

The input dataset, the DI and FEFI simulations, and the output of all experiments shown in Fig 7 can be found on Zenodo at

https://doi.org/ 10.5281/zenodo.14719881 (van den Akker, 2025).

**Author contributions**

TvdA designed and executed the main experiments and the sensitivity analysis. WHL and GRL developed CISM and helped configure the model for the experiments. RSWvdW and WJvdB provided guidance and feedback. TvdA prepared the

manuscript, with contributions from all authors.

**Financial support**

TvdA received funding from the NPP programme of the NWO. WHL and GRL were supported by the NSF National Center for Atmospheric Research, which is a major facility sponsored by the National Science Foundation (NSF) under Cooperative

Agreement no. 1852977. Computing and data storage resources for CISM simulations, including the Derecho supercomputer (https://doi.org/10.5065/D6RX99HX), were provided by the Computational and Information Systems Laboratory (CISL) at NSF NCAR. GRL received additional support from NSF grant no. 2045075.

**Competing interests**

The contact author has declared that none of the authors has any competing interests.

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
