# Peer review of "Competing processes determine the long-term impact of basal friction parameterizations for Antarctic mass loss"

_EGUsphere, 2025_

## Referee Comment (RC2)

**Review of van den Akker et al, TC MS egusphere-2025-441**

This paper presents modelling experiments that explore the 1000/2000-year simulations of Antarctic Ice dynamics under various laws the relate basal friction to sliding velocity and effective pressure. The CISM model used is well -known and has been tested through numerous community benchmarks. It uses a nudging scheme to specify ice thickness, basal friction parameters and other hard to obtain parameters, which looks to work well in general. The overall conclusion is that the gross outcome in terms of sea level rise can be independent of the choice of basal friction law in some circumstances, but strongly dependent under others. I think the authors are correct to reach this conclusion, but have some reservations about some of their simulations.

**General comments**

**The 'DI' experiments are credible and can support the main conclusions, but I think the 'FEFI' experiments are not publishable (yet).**

**DI experiments.** These are the 'standard' simulations, using the mature/ well-known tuning methods associated with CISM, where a basal friction coefficient $C(x,y)$ for each friction law is estimated b to bring the ice sheet thickness into line with observations, and to avoid drift. Although these produce similar VAF(t) for each friction laws, the authors demonstrate that this is a case where differing dynamics adding up to quite similar gross outcomes. Fig 10 in particular shows that the Zoet-Iverson Coulomb limited friction law simulations involve more buttressing (and less basal friction), and the Power law simulations show the opposite behaviour. No doubt with sufficiently high (unrealistic>?) melt rates, the Zoet-Iverson simulations could be denied the buttressing too, and the gross outcome might then differ more. At this point, the authors can conclude that the choice of friction **does** matter (but you need to look at the detail to see that)

**FEFI experiments.** These use a modified / novel tuning method, where an additional flow enhancement factor E(x,y) is estimated, to bring the model in line with observed velocity in additional to the DI constraints (where E = 1). This is a good idea, and indeed many groups find they need to estimate $E(x,y)$ at least in ice shelves. As the authors note, this brings a new level of underdetermination to the estimation problem, and scepticism about the results is required. So far, I have no objection.

However, the process here differs from more typical cases in that it is sensitive to the velocity, rather than to horizontal parts of the strain rate. As a result, the interaction between $C$ and $E$ in the tuning process is different and appears to have produced (as the authors note) radical results that are at odds with convention. Nothing wrong with that, but looking at Fig 6, the outcomes are difficult to accept. The enhancement factor itself shows blocks of much reduced / much increased effective viscosity, and that results in (for example) flow which is dominated by SIA-like internal deformation of ice in the trunk of Thwaites glacier. It is true that some authors find that the SSA is inadequate here, but SIA would be a volte-face.

To be fair to the authors, they are not claiming that their FEFI results are plausible, just they demonstrate sensitivity to underdetermined parameters. Hard to disagree! But a primary result in a glaciology (as opposed to an inverse problem paper) should be citable, and I don't think we would be happy to see future papers citing this paper as evidence that Thwaites glacier is well described by SIA

Possibilities to address this?

1. Remove the FEI results and form the natural conclusions from the DI results. Place the FEFI results in a supplement and make a note in the main text.

2. Argue that the FEFI *E(x,y)* results are credible (a tall order, but possible)
3. Improve the FEFI procedure so that it does produce credible *E(x,y) (e.g by nudging to match horizontal strain rates rather than surface velocity?) – but I think that could be a second paper*

**Specific Comments**

Abstract: obviously, if the paper is revised as I suggest, the last few lines are only weakly supported and so should not appear in the abstract.

L76 T = \beta u = … . The \beta u is not needed – many models do this as part of their implementation, but don't think \beta is mentioned again.

L95. Eq 1.4 is sometimes called a Budd law.

L114. In all four laws friction increases with speed, with diminishing returns (but tend to a limit in the ZI/Schoof cases)

Figure 1. The asymptotes could be added to the figure.

Eq 1.7 R_f rather than \Chi _f ? \Chi is dimensionless, but \Chi f is a stress (like the R components)

L141. \Chi is not a term

L145. Sorry, I don't see the logic here.

L148. 'Shelf kill'. In the interests of a less macho phrase, how about 'Shelf removal'. I know that shelf kill has been used elsewhere.

155. Purest -> simplest / most direct?

L160 – the whole paragraph refers to something in the supplement, but I don't think you need to further show the utility of these well-know buttressing indicators.

Section 2.3 – this section seems a little disorganised. In particular, the nudging equations are introduced immediately before a general introduction to the nudging approach. The tables could be moved to an appendix since the parameters are usually defined in-text.

L227 – I would like more detail (i.e math expressions) at this point.

L265 – 'other factors' – e.g damage, fabric formation, errors in the temperature field

Fig 4 – could the panels be larger/split up?

Fig 10 – actually a more general comment – the shelf removal figures are the more useful in your text, whereas the buttressing number only helps you (vaguely) reiterate a known point about the two halves of Thwaites ice shelf / buttressing being greater near the GL. I would remove the buttressing number analysis, so that the right-hand panels of fig 10 can be larger.

Fig 11. If you *do* include the FEFI results (I suggest not), then show both DI and FEFI at years 575 and 775 (i.e. four panels).

L580- 600 – clearly would be removed if you remove the FEFI material.

L602 – you could compare your E(x, y) with other model results.

L615 – I don't agree in this case because the other models mentioned don't differ from one another in the same way as the DI and FEFI models differ from one another. I am sure you are correct to say that differing initial conditions could explain any number of discrepancies between models, I just do not think the FEFI/DI contrast is representative.

**Technical Corrections**

Eq1. Use f(x, y) notation to make spatially varying vs constant parameters clear?

80 $\tan \phi$, not $tan \phi$. In a similar vein, there are frequent italic subscripts in equations that should probably be roman {\rm } (e.g in eqns 1 & 2)

L112 'asymptote' is not a verb (usually).

---

## Author Comment (AC1)

**Author reply to Reviewer #1: 'Competing processes determine the long-term impact of basal friction parametrizations for Antarctic mass loss'**

Author comments in blue

Preamble comment, also copied to the author responses to Reviewer #2:

We sincerely thank both reviewers for their constructive feedback and for suggesting valuable and thought-provoking points for improvement. However, the reviewers offer differing interpretations of the FEFI method, its results, and the conclusions drawn from our analysis. In summary, Reviewer #1 views the new FEFI method as a significant contribution to the paper that could be emphasized further but also notes instances where our results may have been overinterpreted. In contrast, Reviewer #2 agrees with the conclusions we reached based on our results but advises caution in how we present the FEFI method and initialization outcomes, warning that others might cite this work as definitive evidence of deformational flow at the Thwaites Glacier grounding line.

We agree with Reviewer #1 that, in parts of the manuscript, we may have overinterpreted our results. In our author response, we identify several sections where we will provide a more thorough description and analysis of the results and will moderate our conclusions. We also agree with Reviewer #2 that we do not wish for our paper to be cited as proof that the Thwaites Glacier grounding line is dominated by deformational flow, and we acknowledge the importance of clearly communicating the level of confidence in our inverted results. While we would like to retain these results in the main text, as suggested by Reviewer #1, we propose to include several critical notes to clarify the interpretation of the inverted parameters.

In this study, the authors explore the influence of basal friction laws and initialisation procedures on projections of Antarctic ice-sheet evolution, with a particular focus on the Amundsen Sea Embayment. To do so, they use the ice sheet model CISM, initialised with two variants of a forward inversion nudging scheme: the default one, for which basal friction coefficients and ocean temperature corrections are tuned to match observed ice thickness from the Bedmachine dataset, and a novel one (called FEFI), in which flow enhancement factors are, in addition to the basal friction and ocean temperature correction, nudged to match observed ice surface velocities. In both cases, the observed dHdt is imposed as an additional term to the mass transport equation during the initialisation, following van den Akker et al. (2025), forcing the model to reproduce the observed trends in forward simulations. Starting from these two distinct initial states, the ice-sheet model is run forward in time for 1000-2000 years under constant present-day climate conditions. To test the sensitivity to the choice of friction law, friction fields from both initial states are rewritten (following Brondex et al., 2017, 2019) to convert the initial state for other friction laws. This allows the authors to run, for each initial state, forward simulations under four widely used basal friction laws while starting from the same geometry.

When extending their 8 simulations in time, the authors note a higher sensitivity to the choice of basal friction law for the FEFI initial state than for the DI one. Based on this, they conclude that the model can be tuned to be sensitive to the choice of basal friction law or not. However, I am not sure that the results presented in the manuscript really support this statement. In section 4.2.1 and with Figures 8-10, the authors do show sensitivity of the DI simulations to the choice of basal friction law, even though less pronounced than for the FEFI case. I think that caution should be made when stating that 'the model can be tuned to be sensitive to the choice of basal friction law or not', as written in the abstract.

We thank the reviewer for their careful consideration of our manuscript and their insightful comments. We will change this sentence, related to our answers on comments made below, to: 'this model, CISM, can be tuned to be more or less sensitive to the choice of basal friction law while matching present-day observations equally well'.

To explain the different sensitivities between the simulations, the authors refer to a connection between buttressing and basal sliding in the ASE. However, I find the interpretation of the buttressing analysis in this study somewhat unclear, and I struggle to understand what the authors are trying to demonstrate with the buttressing factors. Although the abstract mentions strong geometry-driven connections, these are difficult to identify in the results section. In particular, it is not entirely clear to me how the comparisons in Figures 10 and 13 support the proposed geometry-driven connection between buttressing and basal sliding. The buttressing numbers shown on the left-hand side of these figures are primarily influenced by differences in ice-shelf geometry at a given time. Therefore, concluding that "buttressing is stronger for one sliding law than another" may be an overinterpretation. Similarly, the increases in velocity following ice-shelf removal (right-hand side of the figures) are more likely driven by the direct influence of the sliding laws on grounded ice velocity (as clearly illustrated in Figure 1) rather than by differences in the buttressing capacity of the ice shelves. I may be misunderstanding the analysis; if so, the results section should be improved to better clarify the influence of buttressing in the simulations.

What we tried to convey is the following: there are two main processes opposing an initial acceleration of a marine-terminating glacier by a floating ice shelf: 1) an increase in basal friction at the grounded part slows down the glacier and 2) an increase in buttressing can increase the ice shelf size and act as a plug, if the bedrock profile allows for enough pinning points or a confined bay. We tried to demonstrate that for a collapsing Thwaites Glacier (TG), process 1 dominates when using the powerlaw and process 2 when using regularized Coulomb friction. In Figure 10, we tried to demonstrate that the ice shelf that forms when the grounding line of TG recedes is buttressing the upstream flow more when using a regularized Coulomb friction law. For the latter law, the ice shelf is thicker because of the larger grounding line flux. We quantified the buttressing strength in two ways, with the stresses at the grounding line and by doing ABUMIP-experiments where we instantly remove the ice shelf and assess the reaction of the upstream grounded ice. In this Figure 10, both calculations show that the ice shelf that forms with the regularized Coulomb sliding law is stronger (i.e. has a higher buttressing capacity) compared to the simulation with power law sliding.

We agree with the reviewer that stating 'buttressing is stronger for one sliding law than another' is an overinterpretation and that it should be made clear in our manuscript that it only holds for 1) our ice sheet model CISM, 2) this specific geometry of a retreating TG with no forcing other than the present day imbalance and 3) for the specific time period during the collapse as shown in Fig. 10 and Fig. 13.

We also agree with the notion that the increases in ice velocities in the right panels of Fig. 10 are more demonstrating the braking effect of the basal friction law, where we see that the power law sliding brakes the accelerating glacier more. This supports point 1) in the first section of this answer and lets us reframe and summarize our conclusions related to the geometric connection as follows:

'There are two main processes opposing an initial acceleration of a TG grounding line retreat initialized with the present-day imbalance. With regularized Coulomb sliding, the dominant braking force is buttressing, whereas with power law sliding, the dominant braking force is basal friction. This holds for the specific case of TG because of the speed of the collapse combined with the formation of a confined ice shelf following grounding line retreat.

We will change the manuscript at several points to remove the overinterpretation and reshape our result sections, in particular around Figure 10 and lines 375–386 and 420–484. See our detailed answers to the specific comments below.

I am wondering whether the differences in sensitivity observed between simulations initialised with the DI and FEFI methods may partly be explained by the differences in velocity fields between the resulting initial states. While both initialisation methods lead to similar ice sheet geometries that agree well with the observations, the associated velocity patterns differ since the FEFI method allows for a reduction of the misfit with observed ice surface velocities through the tuning of the enhancement

factor. The DI initialisation produces an initial velocity field that overestimates flow speeds in the ASE (as shown in Fig. 4), whereas the FEFI inversion reduces this misfit and yields a 'slower' initial ASE. My guess is that this difference likely plays a role when the model is extended into the future, delaying the onset of collapse for both Thwaites and Pine Island glaciers in the FEFI case. This delayed response could make the ice sheet more sensitive to differences in basal friction. In contrast, for the DI case, the imprinted dHdt trends combined with faster velocities dominate the ASE signal, leading to a fairly rapid collapse of the region. I may be mistaken, but I think this is a possibility worth exploring. It would be helpful if the authors discuss this aspect more explicitly and provide a clearer visualisation of the differences between the two initial states in the ASE. Current figures make this comparison difficult.

We agree with the reviewer that the surface velocity pattern of the FEFI simulation agrees better with observations and that we should discuss this effect in more detail. We note that, as shown in Van Den Akker et al. (2025), the velocities are showing an alternating error pattern for the DI initialization at TG and PIG. PIG has a main flowline which is too slow compared to observations, but margins that flow too fast, and a similar pattern between the central flow line and the marginal zones can be seen for TG. It is true that these ice surface velocity misfits are considerable, but because of the compensating effect across the cross-profile, the grounding line fluxes of both glaciers agree well with observations. We would therefore argue that although the ice surface velocity misfit is relatively large, the grounding line fluxes agree well with observations for the DI simulation. Consequently, our DI simulation is as fit as the FEFI simulation for the purpose of projecting future unforced retreat, and that our DI simulation is not necessarily biased toward excessive retreat.

To provide more detail, we will add zooms of the velocity misfit of the ASE and plots of the grounding line fluxes of TG, PIG and the observations to the supplementary material. We will discuss these grounding line fluxes and velocity misfits in section 4.1.1 and 4.1.2.

We will include the following figures in the supplementary material, showing the velocity misfit between the DI and FEFI zoomed in:

[Figure]

Figure S10. Ice surface velocity misfit for DI (left) and FEFI (right) at the end of initialization. Misfit is shown as model minus observations (of Rignot et al. (2011)). The modelled grounding line is shown in black, and the observed grounding line in green.

We will furthermore include the following figure in the supplementary material showing the velocity and thickness error and integrated grounding line flux across the main flow lines of PIG and TG for both simulations:

[Figure]

Figure S11. Grounding line fluxes (red, dashed: CISM; dotted: observations), thickness differences (blue crosses) and ice surface velocity differences wrt observations (blue dots) of PIG (left) and TG (right) for the default initialization DI. The integrated grounding line fluxes represented as ice velocity times the ice thickness (modelled/observed) are respectively for PIG 30.4/31.1 km2 yr-1, and for TG 24.5/26.1 km2/yr.

[Figure]

Figure S12. Grounding line fluxes (red, dashed: CISM; dotted: observations), thickness differences (blue crosses) and ice surface velocity differences wrt observations (blue dots) of PIG (left) and TG (right) for the default initialization FEFI. The integrated grounding line fluxes represented as ice velocity times the ice thickness (modelled/observed) are for PIG 30.6/31.1 km2 yr-1, and for TG are 25.9/26.1 km2/yr.

We will also add the following paragraph to section 4.1:

'Initializing the model with the DI approach results in a dipole misfit in surface ice velocities at the TG grounding line: the Eastern Thwaites glacier flows too slowly, while the Western Thwaites glacier is too fast (Figure S10). A similar dipole pattern emerges near the PIG grounding line, where the model overestimates velocities along the shear margins but underestimates flow along the main trunk. These discrepancies are likely due to the absence of damage representation in the shear zones, an effect that, if included, would increase the velocity gradient in the shear zone, allowing for an even more sharply defined ice stream bounded by near-stagnant ice (Lhermitte et al., 2020; Izeboud and Lhermitte, 2023).

It is important to note that the integrated grounding line fluxes shown in Figure S11 for the DI and S12 for FEFI are close to observational estimates. This suggests that the slower main flow of PIG and

the faster-flowing shear margins approximately balance out, resulting in a total ice flux that matches observations, an outcome that also holds for TG. Applying FEFI significantly reduces the velocity misfits, though the dipole patterns in ice surface velocity misfit persist because the flow enhancement factor inversion is constrained to a minimum (0.1) and maximum (10) value, which it obtains in these regions (see Fig 6). Nevertheless, the integrated ice fluxes through the main flow lines of both glaciers remain close to observed values, even slightly improved compared to the DI initialization. While we acknowledge the surface velocity errors and their potential influence on unforced simulations, the agreement in ice fluxes supports the use of both initialization methods, DI and FEFI, for future unforced model runs.'

Overall, the main focus of this paper is to demonstrate how initialisation choices strongly constrain modelled behaviour. The implications of the initialisation procedure for how we interpret model sensitivity and the range of future ice-sheet responses have already been shown previously (e.g., Seroussi et al., 2019), and this study brings another piece of evidence to support that. That said, the authors should be careful not to overinterpret their results. Relying solely on an initialisation method that explicitly targets observed surface thinning rates raises concerns about overfitting, especially in the ASE, where trends are significant. It is important that the authors clearly acknowledge the limitations of this approach. To strengthen the analysis, I recommend adding some simulations. For example, the same experiments could be performed, but without incorporating the surface thinning rate (dH/dt) in the mass transport equation. I don't believe this would require too much additional work and could reinforce the analysis.

Again, we agree that we may have overinterpreted our results. We refer to our earlier answer, where we propose to change certain parts of the results, discussion, and conclusion sections to adjust the interpretation. We also agree that the addition of a no-dH/dt simulation can enhance our results, However, including an unforced future simulation without dH/dt will result in no ice sheet change as we then initialize our ice sheet model to be stable (i.e. dH/dt=0) and apply no further forcing. What we can simulate to reinforce our analysis is a FEFI-initialization without dH/dt. In that simulation, we observe an even stronger deformational regime on a large area of the ASE but with a low ice surface velocity bias. We interpret that as follows: when initializing an ice sheet model with the observed ice thickness and ice surface velocities, adding the mass change rates is necessary in regions where the highest thinning rates occur. Not including those (i.e. assuming dH/dt = 0) and forcing the model to produce the ice surface velocities in agreement with observations will make the model try to increase surface velocities through a lower viscosity and to increase the basal friction to prevent the ice thickness from decreasing. In other words, if our momentum balance equations are correct and the observations are perfect, imposing the observed ice thickness and ice surface velocities should lead to the observed pattern of mass change rates. Not including the latter and thereby assuming that they are zero will lead to compensating errors.

We will add a discussion on the no-dH/dt FEFI initialization with the same figures as Figure 4 and Figure 5 in the Supplementary Material. We will add an accompanying Figure S13 (see below), and we will add the following paragraph to section 4.1, ln 386:

'We repeated the FEFI initialization for a simulation without incorporating the present-day mass change rates; this simulation is labelled as no-dH/dt-FEFI. Future projections from this initialisation are not relevant for this study, as an unforced future simulation without the present-day imbalance results in negligible ice mass change and grounding line movements, but the initialization reveals a striking pattern: the ASE region receives a maxed out $C_c$, while ice thickness is still underestimated.

When initializing an ice sheet model using observed ice thickness and surface velocities, it is essential to incorporate present-day mass change rates—particularly in regions experiencing the highest thinning. Omitting these rates (i.e., assuming dH/dt = 0) and still forcing the model to match observed surface velocities leads to compensating behaviour. The model will increase surface velocities by lowering viscosity, and to prevent ice thinning, it will raise basal friction. In essence, if the

momentum balance equations are accurate and the observational data reliable, then prescribing both the observed ice thickness and surface velocities should naturally reproduce the observed mass change pattern. Ignoring the latter and assuming zero mass change forces the model to compensate by introducing systematic errors.'

[Figure]

Figure S13. Modelled Antarctic Ice Sheet initialized state with FEFI and no mass change rates applied. (a) thickness difference with respect to observations (Morlighem et al., 2020). The modelled grounding line is shown in black, and the observed grounding line in green (only visible where it does not overlap with the modelled one). (b) ice surface velocity difference with respect to the observations (Rignot et al., 2011). Positive values indicate regions where CISM overestimates the ice velocities. (c) the inverted $C_c$ from Eq 1.1 using Eq 1.2. and (d) the inverted ocean temperature perturbation under the main shelves.

In terms of presentation, the manuscript would benefit from improved clarity in the figure captions. For several figures (e.g., Figures 9, 10), captions lack some key information, e.g., whether the results are from the DI of FEFI initialisation procedure. The reader needs to search for the information within the text. Efforts should be made to ensure that the captions and figures are as self-explanatory as possible. Similarly, I find the Results section difficult to follow. The storyline lacks clarity, and the key messages are scattered throughout the text, making it difficult to identify the main findings. I recommend reorganising this section to improve its logical flow and highlight the key results more clearly. In addition, I would suggest that the authors put more emphasis on the adjusted FEFI initialisation procedure that they present. I think that this is a key contribution of this study. A method allowing for a reduced misfit for both ice thicknesses and ice surface velocities is very valuable, and it deserves to be more clearly highlighted in the paper.

We will revise and clarify our figure captions. While we appreciate that the reviewer values our new FEFI method, we would argue against posing the FEFI simulation as an advantage/improvement over our existing DI. This will be clarified, in line with the remarks of reviewer 2 who argues that care should be taken with the FEFI results as they are more an artefact of the procedure than a better fit to a more physical result. We will write in our conclusion:

'Despite the risk of overtuning, the FEFI simulation shows a decreased misfit between observed and modelled ice surface velocities. This makes it a promising new additional initialization method next to the DI. More research is necessary to constrain FEFI, possibly with field observations, and methods need to be developed to determine the flow enhancement factor more physically based and advect the inverted properties of FEFI with the ice flow, as it moves away from its initialized position, instead of prescribing it as a function of its location'

Despite these limitations, this is an interesting and valuable study. With revisions that address the concerns raised above and the comments below, I believe the manuscript could be suitable for publication.

We thank the reviewer again for a thorough read and detailed comments.

**Specific comments:**

*Abstract*

l.10: Those two starting sentences are somewhat contradictory. Maybe reformulate as, e.g., 'Previous studies do not agree on the magnitude of the influence of basal friction laws in sea-level projections'.

We thank the reviewer for this suggestion, and we will replace Ln 9-10 with 'Previous studies do not agree on the magnitude of the influence of basal friction laws in sea-level projections.'

l.18: What is meant by underdetermined here? This should be clarified.

The degrees of freedom combined with limited observations. We will rewrite 'the underdetermined nature of ice sheet models' with: 'the degrees of freedom when ice sheet modelling with a considerable amount of parameterizations and modellers choices is combined with sparse observations not able to fully constrain those choices.'

l.18-19: This is quite a strong claim, and I am wondering how constructive it is to make. I think stating that model results are dependent on modelling choices and the initialisation procedure is sufficient. You can indeed tune your model to force it towards a given behaviour, but this does not mean that modelling results are not useful. This underscores the importance of validating model outputs against observations.

We agree with the reviewer and will remove 'The latter makes it difficult to base general claims on ice sheet modelling results'

*Introduction*:

l.42: It can be worth mentioning that the second category of basal friction laws was originally developed to capture the behaviour of sliding over deformable till, in contrast to the first category, which typically represents sliding over hard beds.

We will add to L42: '.. and were originally developed to represent basal sliding over hard bedrock' and to L43: '.. and were originally developed to represent sliding over softer, deformable till.'

l.49: 'If the thicker ice shelf persists' – This implies that increased grounding line flux leads to a thicker ice shelf, which I am not sure is necessarily the case. Buttressing capacity is also simply influenced by changes in the shape/geometry of the ice shelf.

We will rewrite L49: If the thicker ice shelf persists and becomes more heavily stuck within the bedrock profile, the buttressing capacity of the ice shelf increases and opposes the initial acceleration of the inland ice'.

l.64: 'showing the importance of ice-shelf buttressing' – I leave it up to the authors whether they want to keep this or not, but this may feel out of place, as it comments on a result while still stating the study's goals.

We will remove this subsentence

l.71-72: I am not entirely convinced by the reasoning presented here. I am not sure that having a compensating effect from buttressing means that the ice sheet evolution is not sensitive to the choice of the basal friction law. These are two distinct processes which both contribute to the ice dynamics. The combination of both leads to the ice-sheet response.

We will rewrite this as 'Two key processes govern the backstress experienced by a glacier: buttressing and basal friction. Which of these dominates depends on the glacier's geometry during collapse. We present a specific case in which a reduction in basal friction—caused by adopting a different basal sliding law—is offset by a corresponding increase in buttressing. Whether this compensation occurs ultimately depends on the glacier's geometry, which we show is itself shaped by specific choices made during the initialization phase.'

*Methods*

l.130: Could you clarify what is meant by 'linearised' stress balance here?

Normalized in the direction of the ice flow. However, this is confusing, so we will remove 'linearised'

l.133: Which Eq. 1 are you referring to?

Eq 1in the Supplementary material of Fürst et al. (2016). We will add this.

l.162-163: Maybe specify that both numbers correlate well for the theoretical case of MISMIP+ but show low correlation when applied to Antarctica.

We will add to L163: 'little to no correlation could be found when the same method was applied to AIS ice shelves, as shown in Fig S9.'

l.182: For a smoother transition, briefly remind what Cc/Cp is.

We will add to L182: 'Since the spatially varying parameters $C_c$ and $C_p$ in Eq 1.1 – 1.4 are poorly constrained by theory and observations … '

l.186: So if I understand correctly, you aim for an a priori field of Cc. Can you comment on the influence of r?

We will add to L186 'Where κ is the relaxation timescale and r a parameter controlling the strength of the relaxation term compared to the other terms.

l.187-189: How sensitive are the results to the choice of Cr? It would be interesting to comment on this.

Since $C_c$ and $C_r$ are poorly constrained, any field of $C_r$ based on bedrock elevation height can be chosen. Answering this interesting point of the reviewer therefore requires several new initializations with different fields of $C_r$. We did not do a sensitivity analysis for $C_r$, since it falls outside the scope of this study.

l.196: It is the ocean temperature which is tuned, isn't it? I suggest clarifying this in the sentence.

Yes, correct. We will change 'The basal melt rates' to 'The ocean temperature'.

l.200: Do I understand correctly that you are not applying the temperature corrections per basin provided in Jourdain et al. 2020, but rather calculate it yourself locally through this nudging procedure?

Yes, we use our own inverted values.

l.205: What is the initial value attributed to the enhancement factor when starting the inversion? And what is the default value used in the default initialisation?

We use values of 1 for grounded ice and 0.5 for floating ice, based on earlier studies with CISM on the present-day AIS (Lipscomb et al., 2021; Van Den Akker et al., 2025). We will add to L207: 'Following Lipscomb et al. (2021); Van Den Akker et al. (2025), when not inverting $E$ a value of '1' is used for grounding ice and 0.5 for floating ice. These values are also the initial values used when starting an inversion using Eq 1.15.

l.229-230: Can you specify how exactly you define such a conflict?

We will add to L229: (e.g. when modelled ice is too slow and too thin, the basal friction inversion will try to increase the ice thickness by decreasing ice velocities, and the flow enhancement factor inversion will try to increase the surface velocities).

l.252: Refer to the figure showing the comparison between modelled and observed velocities here.

We will add to the end of L252: 'see section 4.1 and Figures 4 and 5.'

l.255: Which version of Bedmachine are you using?

Version 1 from Morlighem et al. (2020), we will add 'version 1' to L255

l.273: For clarity, please remind what those datasets are.

We will add to L273: '(ice thickness, ice surface velocities, and the mass change rates)'

l.279: Please clarify what could be such a physically implausible behaviour.

We will add to L279 '(e.g., deformation-driven flow in fast-flowing regions close to the grounding lines, rather than sliding-dominated flow)'

l.300: This seems to be missing from the supplementary material.

We will add the following to the Supplementary materials:

Rewriting of the basal friction parameterizations

For Eq 1.7, the rewriting is straightforward:

$$C_p = \frac{C_c N \left(\frac{u_b}{u_b + u_0}\right)^m}{u_b^m} \tag{1.21}$$

For the Pseudoplastic law, a similar method is used:

$$C_{cp} = \frac{C_c \left(\frac{u_b}{u_b + u_0}\right)^m}{\left(\frac{u_b}{u_0}\right)^{\frac{1}{pp}}} \tag{1.22}$$

Here, $C_c$ is the inverted free parameter in Eq 1.3, and $C_p$ the constant needed in Eq 1.7, $C_{cp}$ the constant needed in Eq. 1.8. Rewriting Eq. 1.8 is less straight-forward. First we define:

$$\beta = \frac{\tau_{b,Reg}}{u_b} = C_c N \frac{u_b^{m-1}}{(u_b + u_0)^m} \tag{1.23}$$

Eq 1.6 has two free parameters, $C_p$ and $C_c$. The former controls the strength of the Weertman style friction, and the latter controls the strength of the Coulomb-style friction. Since we inverted for $C_c$ in Eq 1.3 in the initialization procedures, we use the inferred value to compute a $C_p$ so that Eq 1.8. gives the same basal friction at the end of the initialization as Eq 1.3:

$$C_p = \left(\frac{\alpha}{1 - \gamma}\right)^{\frac{1}{m}} \tag{1.24}$$

With:

$$\alpha = \left(\frac{\beta}{u_b^{m-1}}\right)^{\frac{1}{m}} \tag{1.25}$$

And:

$$\gamma = \left(\frac{\beta}{C_c N u_b^{m-1}}\right)^{\frac{1}{m}} u_b \quad (1.26)$$

l.302: I don't believe you specified the spatial resolution used for your simulations anywhere. This is important information that should be included somewhere in the methods section.

We added to L169: 'All simulations in this study were done on a 4-km grid'.

*Results*

Figure 4-5: It would be interesting to see the inverted ocean temperature perturbation field for the entire ice sheet. It could be included in the supplementary material while still providing an opportunity to visualise it. In contrast, since your focus is on the ASE, it would be useful to illustrate the influence of both initialisation procedures more clearly. One way to do so would be to include a figure that compares maps of velocity and thickness misfits zoomed into the ASE region and placed side by side. Currently, the need to switch between figures and zoom in makes direct comparison difficult.

We will add the inverted ocean temperature perturbation of the whole ice sheet to the supplementary material, and refer to this figure in L332 and L353. We will include zoom-ins of the velocity misfit in the ASE region including a discussion on the grounding line flux as mentioned as response to one of the major comments above.

Figure 6: Similarly, I would be interested to see the resulting enhancement factors for the whole ice sheet.

We will add a figure like Fig 6a, showing the inverted flow enhancement factor of the whole ice sheet, to the supplementary material.

l.355-356: I don't fully understand why the thickness bias in the EAIS interior has grown compared to the DI. It could be worth commenting on this.

We agree, and note now that the original statement is not precise enough. The thickness bias increases at outlet glaciers of the EAIS, where the flow enhancement factor inversion apparently tries to erase velocity errors at the cost of the thickness bias. We will change L355-356 to:

l.356-357: While the thickness bias in the interior of the EAIS has persisted, the bias in the outlet glaciers has increased in magnitude. This suggests that either the prescribed surface mass balance (SMB) is too low, or that the inversion of the flow enhancement factor has led to increased surface velocities. This effect is evident in the inverted Coulomb C field: in the FEFI simulation, outlet glaciers in Dronning Maud Land all reach a value of '1'. At this point, the friction inversion can no longer counteract ice loss, and the enhanced flow speeds increase ice flux, thinning the ice and further amplifying the thickness bias.

l.360: How do you explain the fact that the thickness misfit in the shelves is larger with FEFI than DI?

The thickness misfit increases slightly on the shelves and is mainly negative. The modelled ice shelves are too thin, likely because the flow enhancement factor increases their velocities and thereby the ice flux to the calving front. The ocean temperature inversion can counteract this only until the basal melt rates are 0, e.g. when the inverted ocean temperature perturbations is equal to minus the thermal forcing. If then the flow enhancement factor increases the velocities further, the ice shelves will thin.

We will add to L362: The thickness misfit over the ice shelves increases slightly and is predominantly negative, indicating that the modeled shelves are generally too thin. This is likely due to the flow enhancement factor increasing ice velocities, which in turn raises the flux toward the calving front. While the ocean temperature inversion can compensate for this thinning by reducing basal melt rates, it is limited—once the inverted ocean temperature perturbation equals the negative of the thermal forcing, basal melt cannot decrease further. If the flow enhancement factor continues to accelerate ice flow beyond this point, the shelves will continue to thin.

Figure 6: I am not sure what Figure 6b represents. Is it the change in velocity compared to DI? Or is it a zoom on the velocity misfit with respect to the observations? The caption should be clarified. If it is the change in velocity compared to DI, then a zoom on the velocity misfit with respect to the observations would be useful.

This is the velocity difference change in percentage compared to the DI. We will add this to the colorbar label and to the caption, and we will include zooms of the ASE of the velocity misfit in both inversions as a respond to an earlier comment.

l.384-386: I am not sure what you mean to say by this statement. Wouldn't the same be true for DI initialisation?

Yes, the same general statement can be made for DI but not specifically for deformation at the TG grounding line, which does not happen in the DI case.

We will add to L386 'The same general statement can be made for the DI initialization and its inverted parameters, but that inversion did not result in deformation-driven flow at the TG grounding line'

l.413-415: One could argue that the same is true for the simulations starting from DI, even though it is less pronounced.

We mention in L416 'the rate of glacier collapse is much less affected by the choice of the basal friction parameterization'. We think this is sufficiently different from saying 'the DI simulation is not at all sensitive to changes in basal sliding law'.

l.426-427 Isn't this the case for PIG and not TG? Overall, I don't find the results description in this paragraph to be very straightforward. It would be good to try to improve it. It is not obvious to me from Figure 8 that the grounding lines of the power law and PP law and the schoof and ZI laws are grouped together.

We will rewrite this paragraph (L420 – L429) as:

We begin by discussing the results from the DI experiments (solid lines in Fig. 7), where the collapse rate shows little sensitivity to the choice of basal friction law. Figure 8 presents elevation profiles of PIG (top row) and TG (bottom row) along the cross-sections indicated in Fig. 3. Before the collapse (year 250, first column), differences in ice sheet geometry and grounding line position across the four sliding laws are minimal for both glaciers. The final states after collapse (year 750, last column) also appear similar. However, at the midpoint of the collapse (year 500, middle column), the glacier shapes diverge for TG (panel b). For example, in the simulation using power law sliding, the grounding line has retreated roughly 75 km farther than in the Zoet-Iverson sliding case. Notably, the ice upstream of the grounding line is thinner for the Zoet-Iverson sliding case compared to the power law, but the latter shows more grounding line retreat. The Zoet-Iverson sliding case loses more mass further inland, and the powerlaw sliding case loses mass close to the grounding line and has a thinner ice shelf.

l.424-428: So, it seems that the results of the DI experiments are still fairly sensitive to the choice of the basal friction law. Is that correct?

In terms of grounding line position and ice sheet geometry, yes. In terms of sea level rise and integrated ice mass loss (Figure 7), no. We add to the rewritten paragraph of the previous comment the following sentence: 'From Fig 7 we conclude that the integrated mass loss and SLR from the DI sliding law simulations is very similar, but from Fig 8 we see different geometries during the TG collapse.

Figure 8-9: Specify in the captions that those results are for the DI initial state.

We will add 'DI' to the captions.

l.433: I'd suggest removing the 'much'. The GL retreat seems roughly similar, even though slightly more retreated for the power law. I would also say that retreat is slightly more pronounced for the ZI law at year 750 than for the power law.

We will remove 'much'

Figure 10: I'd suggest specifying that the area shown is not the same for the buttressing number and the absolute velocity increase. I was initially a little confused. You could also mark the zoomed area, or simply show the same area for both to make it easier to compare.

We did this because the relevant area for the buttressing number is smaller compared to the area shown for the acceleration number. Making them the same size sacrifices detail. We will add in the caption 'Note the different zooms of the four panels on the left and the four panels on the right. This is done to preserve detail in the left four panels'

l.468-469: I am not sure that I agree with this statement. Due to the different geometries at year 500, it is difficult to attribute differences in buttressing to friction laws. Instead, I would attribute those differences to the evolving geometries themselves, which are, of course, influenced by the friction laws. The analysis is more informative at year 250, when the geometries are still comparable. In this case, I find that the pattern of the buttressing numbers at year 250 is very similar for both friction laws. The velocity increase differs, but it seems to me that this difference results from the friction law's influence on the ice flow response, rather than from differences in the buttressing effect itself. In other words, the buttressing is similar, but the response to loss of buttressing is highly dependent on the friction law, as expected. Overall, I am not convinced that these acceleration numbers provide a valid approach to quantify or isolate a potential influence of basal friction on the buttressing capacity.

We agree with the statement on changing 'attributing the differences in buttressing to the friction laws', to attributing changes in buttressing to the geometry of the ice shelf, which is determined by the friction law. As mentioned in our reply to the second major comment: we agree that we overinterpreted our results. We will rewrite L465-L472 as:

'The right side of Figure 10 shows changes in ice velocity during the shelf-removal experiments at years 250 and 500, comparing Zoet-Iverson sliding (left column) and power law sliding (right column). Following the removal of the ice shelves, grounded ice in the Zoet-Iverson simulation accelerates rapidly—especially at the Thwaites Glacier (TG) grounding line and farther inland—much more so than in the power law case. Two primary mechanisms can slow down a retreating marine-terminating glacier like TG: buttressing and basal friction. At year 500, buttressing at the TG grounding line is stronger in the Zoet-Iverson simulation than in the power law case, as indicated by the higher buttressing numbers. Despite this, the acceleration response to shelf loss (right four panels of Fig. 10) is greater in the Zoet-Iverson case. This is because, in the power law case, basal friction increases with velocity (as shown in Fig. 1), limiting the glacier's speed-up. In contrast, the Zoet-

Iverson simulation exhibits less frictional resistance and thus stronger acceleration. We conclude that, during the TG collapse, buttressing is the primary braking mechanism in the Zoet-Iverson case, whereas increased basal friction dominates in the power law case. Interestingly, due to the specific bed geometry of TG and the conditions in the unforced simulation, both scenarios produce a similar contribution to global mean sea level rise—though driven by different mechanisms. '

l.486: Again, I think that this is quite a strong statement. I agree that Figure 7 showing the evolution of the volume above flotation shows less sensitivity to the choice of friction law with DI than with FEFI. However, you just showed that results with DI are sensitive to the choice of friction law.

We will add to L486 'in terms of integrated ice mass loss and GSML rise contribution.'

l.487: To which glacier collapse are you referring? I'm assuming Thwaites. In lines 407-411, you referred to three stages of retreat for those experiments. Am I correct in understanding that the PIG collapse is triggered after 600 years in all simulations? This should be clarified. It would be interesting to show the pattern of mass loss for all simulations, which could be included in the supplementary materials. Since you don't have many simulations, this sounds feasible and would help clarify the interpretation of the results.

We are referring to the total ASE collapse, which can either start at TG or PIG. PIG collapse is triggered after 600 years indeed, but in the DI simulations because of 'ice piracy' of a collapsing TG and in the FEFI simulations because of its own GL retreat.

We will rewrite L486-489 as:

Unlike the default experiments, the simulations initialized with FEFI exhibit a strong sensitivity to the choice of basal friction law. Also, the collapse of the ASE occurs later, beginning around year 800 in the Zoet-Iverson case and around year 1200 in the power law case, and follows a different pattern. Instead of transitioning directly from present-day-like mass loss rates (~0.3 mm/yr) to full collapse rates (~3 mm/yr), there is an intermediate phase with mass loss rates of approximately 1–2 mm/yr. This transitional phase is driven primarily by the collapse of PIG, which occurs independently of TG in the FEFI simulations.

l.491: I think you mean Fig. 7?

We mean Figure 11.

l.492: What experiment is this for? Based on the DI results presented above, it seemed like the TG and PG collapsed fairly simultaneously, didn't they? Fig. 12 also seems to show this.

For the Zoet-Iverson sliding law. We will add this to L491 and to the caption, see next reply.

Figure 11: What basal friction law is shown on this figure? This should be clarified in the caption.

This is for the Zoet-Iverson experiment, but similar for the powerlaw simulation. We will add to the caption 'Results shown here are from the Zoet-Iverson sliding simulations, but results are similar for the power law simulation'.

l.503: PIG collapse slower than TG – Is this only for the FEFI inversion? The timing of collapse between TG and PIG seems rather equivalent in the DI case.

This is the case in both our simulations but harder to disentangle (other than visually) because of the broader collapse in the DI (TG collapses and draws PIG quickly with it).

Figure 13: I think this is the other way around, as described in the caption. Also, to make sure I understand correctly, is the speedup in the bottom row observed in just one model timestep? The positions of the grounding lines between the top and bottom rows seem quite different. Did this occur in only one timestep? Also, please clarify in the caption what model year is represented in the figure.

Correct, thanks for pointing that out, they are in reverse order. Yes, this is a single timestep, which in our case is 1/6 year (2 months). Rates of grounding line retreat are much higher than the ~2 km year retreat observed today, so in a single timestep and especially during a shelf kill experiment, the grounding line can move visibly. The model year is 775. We will rewrite the caption as follows:

Ice shelf removal experiments and buttressing quantification during the FEFI continuation simulations. The top row shows buttressing values for the Zoet-Iverson (left) and power law (right) cases, while the bottom row displays ice velocity accelerations resulting from the shelf-kill experiments. Note that the grounding line positions differ slightly between the top and bottom rows, despite only a single model timestep (2 months) separating them. This discrepancy is due to the rapid grounding line retreat occurring during the collapse.

l.522: hardly matters – I find this to be a rather strong statement. I agree that a more pronounced speedup is seen with the ZI sliding law, but it also seems like grounding line retreat has been triggered within one timestep in response to the killing of the ice shelves.

We will write instead 'matters less'.

l.522-523: I do not see how the difference in acceleration is smaller for FEFI than for DI. Figures 13 and 10 seem fairly comparable to me. Am I missing something?

We are referring to the strong acceleration of TG's main flow line when removing the shelf in the DI Zoet-Iverson simulation. We do not observe such far inland reaching accelerations in Fig 13. We will remove the word 'much' from L523, and add 'when comparing the size and inland extent of the acceleration number in the main flow line of TG with the effect in Figure 13 around the GL of PIG'.

Section 4.3: I am not convinced by the added value of this section. Stating that the integrated basal melt flux is nearly equal to the grounding line flux for TG seems like an overinterpretation to me. This may be true for the FEFI case, but only prior to the collapse. The same could be said for the ZI-DI case in PIG, though. Therefore, I am having difficulty identifying a key message in Figure 14. Perhaps I have misunderstood the section. If so, it is worth improving its clarity.

We agree with the reviewer that this section is out of place and does not add much to the storyline. We will remove L529 – L539 and Figure 14.

*Discussion*

l.552: It would be interesting to include a statement in the discussion, either in this paragraph or elsewhere, about the role of subglacial hydrology and also the influence of meltwater in the grounding zone (e.g. Hewitt and Bradley). The current simple representation of effective pressure surely influences the ice-sheet response highlighted in this study.

We agree and will add to the discussion the following paragraph:

This study accounts for subglacial hydrology only in a simplified way. We parameterize the effective pressure $N$ according to Leguy et al. (2014), where we assume that $N$ is reduced near grounding lines because of a connection between the subglacial hydrology network and the ocean. While this captures some aspects of including a subglacial hydrological system (e.g. lowering basal friction in areas close

to grounding lines), it does not simulate a complex hydrological network as was done, for example, by Kazmierczak et al. (2024) or Bradley and Hewitt (2024). Including a more complex hydrological network and coupling it via the effective pressure to the basal friction would likely alter our results.

l.553: It is worth noting that both Barnes and Gudmundsson and Wernecke et al. focus on much shorter timescales (decadal). At similar timescales, all of your simulations would likely agree with the results of those two studies, probably as a consequence of your initialisation procedure using present-day dhdt rates.

We will add to L553: 'although the latter two studies focus on shorter timescales (~100 years) and feature much less grounding line retreat than our study (e.g., no WAIS collapse)'.

l.579-583: It is important to emphasise the influence of the initialisation procedure, in which the model is forced to imprint present-day DHDT trends. First, extending this present-day trend into the future dominates your simulation's response on shorter timescales (decadal to centennial). It is only once a significant grounding line retreat is triggered in your simulations, at different times due to differences in the initial states, that the influence of the basal friction laws kicks in. To what extent do you think your conclusions apply to this particular inversion procedure? It is likely that the basal friction field has been overfitted to match the observed dHdt. One alternative would be to apply the nudging scheme to an earlier ice sheet state and then assess whether the model can reproduce historical trends. In that case, different parameter fields might produce valid initial states but result in different model behaviours, highlighting how the initialisation procedure constrains the simulations. Since the observed dHdt has been imposed on the model, it is difficult to determine which parameter values would improve the match to observations. Tuning the basal friction field to match present-day trends prevents testing its ability to capture changes in mass balance in response to varying boundary conditions.

These are valid, and interesting, points! See our reply in a rewritten L579-583 below:

In this study, we use two different initializations to demonstrate that the sensitivity of modeled ice mass loss in the Amundsen Sea Embayment (ASE) to the choice of basal friction parameterization can vary significantly. One initialization exhibits a greater sensitivity than the other, suggesting that initialization plays a key role in determining how strongly ice loss responds to different friction laws. This finding represents a first step toward establishing that sensitivity to basal friction parameterization is itself dependent on the initialization approach. However, our analysis is based on a single ice sheet model (CISM) and only two initialization methods. Furthermore, because the simulations apply only observed present-day mass change rates without additional forcing, the first several centuries primarily reflect a continuation of current retreat trends. It is only after about 500 years that ice mass loss accelerates and differences between the initializations become apparent.

When future ocean warming is applied, the resulting ice shelves in the ASE are expected to be much smaller. As a result, the buttressing effect that moderates retreat in the Zoet-Iverson simulation would be reduced, likely leading to greater projected global mean sea level (GMSL) rise compared to the power law case. Therefore, the results presented here are specific to the CISM model, the initialization techniques used to reproduce present-day mass loss, and the absence of any future forcing. Follow-up studies are needed to evaluate whether our conclusions hold under different scenarios, such as schematic ocean warming or through transient calibrations designed to reproduce historical mass loss trends (e.g. Goldberg et al., 2015)

l.597-598: It is worth mentioning that the FEFI initialisation reduces the overestimation of observed surface velocities in the ASE. This could be a key factor in explaining why the FEFI initial state experiments showed delayed retreat compared to the DI ones. This delayed retreat leaves more room for the influence of basal friction on the rate of GL retreat. In contrast, for the DI initial state, the

response is dominated by the imprinted fast flow in the ASE, leaving less room for the basal friction law.

We will add a discussion on the velocity misfit and the grounding line flux correctness in section 4.1 (see an earlier comment). We will furthermore add to L598:

Also, the FEFI initialization reduces the misfit between modelled and observed ice surface velocities in the ASE, which may partly explain why experiments starting from the FEFI state show a delayed retreat compared to those initialized with DI. This delayed retreat allows basal friction to play a greater role in controlling the rate of grounding line retreat.

*Conclusion*

l.616-618: It would be interesting to discuss potential ways to address this issue. For example, by including initial state uncertainty in ensembles of simulations?

We will add: 'A potential way to address this issue is through standardized tests (e.g. MISMIP+ for different sliding laws) after major changes have been made to the initialization procedure of the ice sheet model. Also, projections could start from an ensemble of many different initializations, all done with different model choices (e.g., inverting for the flow enhancement factor or not, as was done in this study).

**Technical corrections:**

l.27: double spacing at the end of the line: '(TG),  is' We will remove the double space

l.34: 'Sources of modelled ice sheet uncertainty' – replace by 'Sources of uncertainty in modelled ice sheet behaviour' ? Great suggestion, we will replace this

l.56: consistency in citation formatting: 'e.g., Brondex et al., 2017; Sun et al., 2020, Brondex et al., 2019)'. This is the case at several places in the manuscript. We will go through the citations and make sure they are consistent and in the right order

l.58: same: (ABUMIP; Sun et al., 2020) Thanks for pointing this out

l.93: 'Eq 1.4 and is referred to' – remove 'and' we will remove 'and'

l.382: 'depending on' we will change 'dependent' to 'depending'

Figure 9: mix up between bottom and top row for Power Law / ZI. We will switch these.

**References**

Bradley, A. T. and Hewitt, I. J.: Tipping point in ice-sheet grounding-zone melting due to ocean water intrusion, Nature Geoscience, 17, 631-637, 2024.
Fürst, J. J., Durand, G., Gillet-Chaulet, F., Tavard, L., Rankl, M., Braun, M., and Gagliardini, O.: The safety band of Antarctic ice shelves, Nature Climate Change, 6, 479-482, 2016.

Goldberg, D. N., Heimbach, P., Joughin, I., and Smith, B.: Committed retreat of Smith, Pope, and Kohler Glaciers over the next 30 years inferred by transient model calibration, The Cryosphere, 9, 2429-2446, 10.5194/tc-9-2429-2015, 2015.

Izeboud, M. and Lhermitte, S.: Damage detection on antarctic ice shelves using the normalised radon transform, Remote Sensing of Environment, 284, 113359, https://doi.org/10.1016/j.rse.2022.113359, 2023.

Kazmierczak, E., Gregov, T., Coulon, V., and Pattyn, F.: A fast and simplified subglacial hydrological model for the Antarctic Ice Sheet and outlet glaciers, The Cryosphere, 18, 5887-5911, 2024.

Leguy, G., Asay-Davis, X., and Lipscomb, W.: Parameterization of basal friction near grounding lines in a one-dimensional ice sheet model, The Cryosphere, 8, 1239-1259, 2014.

Lhermitte, S., Sun, S., Shuman, C., Wouters, B., Pattyn, F., Wuite, J., Berthier, E., and Nagler, T.: Damage accelerates ice shelf instability and mass loss in Amundsen Sea Embayment, Proceedings of the National Academy of Sciences, 117, 24735-24741, 2020.

Lipscomb, W. H., Leguy, G. R., Jourdain, N. C., Asay-Davis, X., Seroussi, H., and Nowicki, S.: ISMIP6-based projections of ocean-forced Antarctic Ice Sheet evolution using the Community Ice Sheet Model, The Cryosphere, 15, 633-661, 2021.

Morlighem, M., Rignot, E., Binder, T., Blankenship, D., Drews, R., Eagles, G., Eisen, O., Ferraccioli, F., Forsberg, R., and Fretwell, P.: Deep glacial troughs and stabilizing ridges unveiled beneath the margins of the Antarctic ice sheet, Nature Geoscience, 13, 132-137, 2020.

Rignot, E., Mouginot, J., and Scheuchl, B.: Ice flow of the Antarctic ice sheet, Science, 333, 1427-1430, 2011.

van den Akker, T., Lipscomb, W. H., Leguy, G. R., Bernales, J., Berends, C. J., van de Berg, W. J., and van de Wal, R. S.: Present-day mass loss rates are a precursor for West Antarctic Ice Sheet collapse, The Cryosphere, 19, 283-301, 2025.

---

## Author Comment (AC2)

**Author reply to Reviewer #2: 'Competing processes determine the long-term impact of basal friction parametrizations for Antarctic mass loss'**

Author comments in blue

Preamble comment, also copied to the author responses to Reviewer #2:

We sincerely thank both reviewers for their constructive feedback and for suggesting valuable and thought-provoking points for improvement. However, the reviewers offer differing interpretations of the FEFI method, its results, and the conclusions drawn from our analysis. In summary, Reviewer #1 views the new FEFI method as a significant contribution to the paper that could be emphasized further but also notes instances where our results may have been overinterpreted. In contrast, Reviewer #2 agrees with the conclusions we reached based on our results but advises caution in how we present the FEFI method and initialization outcomes, warning that others might cite this work as definitive evidence of deformational flow at the Thwaites Glacier grounding line.

We agree with Reviewer #1 that, in parts of the manuscript, we may have overinterpreted our results. In our author response, we identify several sections where we will provide a more thorough description and analysis of the results and will moderate our conclusions. We also agree with Reviewer #2 that we do not wish for our paper to be cited as proof that the Thwaites Glacier grounding line is dominated by deformational flow, and we acknowledge the importance of clearly communicating the level of confidence in our inverted results. While we would like to retain these results in the main text, as suggested by Reviewer #1, we propose to include several critical notes to clarify the interpretation of the inverted parameters.

This paper presents modelling experiments that explore the 1000/2000-year simulations of Antarctic Ice dynamics under various laws the relate basal friction to sliding velocity and effective pressure. The CISM model used is well -known and has been tested through numerous community benchmarks. It uses a nudging scheme to specify ice thickness, basal friction parameters and other hard to obtain parameters, which looks to work well in general. The overall conclusion is that the gross outcome in terms of sea level rise can be independent of the choice of basal friction law in some circumstances, but strongly dependent under others. I think the authors are correct to reach this conclusion, but have some reservations about some of their simulations.

**General comments**

**The 'DI' experiments are credible and can support the main conclusions, but I think the 'FEFI' experiments are not publishable (yet).**

**DI experiments.** These are the 'standard' simulations, using the mature/ well-known tuning methods associated with CISM, where a basal friction coefficient C(x,y) for each friction law is estimated b to bring the ice sheet thickness into line with observations, and to avoid drift. Although these produce similar VAF(t) for each friction laws, the authors demonstrate that this is a case where differing dynamics adding up to quite similar gross outcomes. Fig 10 in particular shows that the Zoet-Iverson Coulomb limited friction law simulations involve more buttressing (and less basal friction), and the Power law simulations show the opposite behaviour. No doubt with sufficiently high (unrealistic>?) melt rates, the Zoet-Iverson simulations could be denied the buttressing too, and the gross outcome might then differ more. At this point, the authors can conclude that the choice of friction does matter (but you need to look at the detail to see that)

**FEFI experiments.** These use a modified / novel tuning method, where an additional flow enhancement factor E(x,y) is estimated, to bring the model in line with observed velocity in additional to the DI constraints (where E = 1). This is a good idea, and indeed many groups find they need to

estimate E(x,y) at least in ice shelves. As the authors note, this brings a new level of underdetermination to the estimation problem, and scepticism about the results is required. So far, I have no objection.

However, the process here differs from more typical cases in that it is sensitive to the velocity, rather than to horizontal parts of the strain rate. As a result, the interaction between C and E in the tuning process is different and appears to have produced (as the authors note) radical results that are at odds with convention. Nothing wrong with that, but looking at Fig 6, the outcomes are difficult to accept. The enhancement factor itself shows blocks of much reduced / much increased effective viscosity, and that results in (for example) flow which is dominated by SIA-like internal deformation of ice in the trunk of Thwaites glacier. It is true that some authors find that the SSA is inadequate here, but SIA would be a volte-face.

To be fair to the authors, they are not claiming that their FEFI results are plausible, just they demonstrate sensitivity to underdetermined parameters. Hard to disagree! But a primary result in a glaciology (as opposed to an inverse problem paper) should be citable, and I don't think we would be happy to see future papers citing this paper as evidence that Thwaites glacier is well described by SIA Possibilities to address this?

1. Remove the FEFI results and form the natural conclusions from the DI results. Place the FEFI results in a supplement and make a note in the main text.

2. Argue that the FEFI E(x,y) results are credible (a tall order, but possible)

3. Improve the FEFI procedure so that it does produce credible E(x,y) (e.g by nudging to match horizontal strain rates rather than surface velocity?) – but I think that could be a second paper

We thank the reviewer for raising this important and thought-provoking discussion, for offering constructive suggestions on how to proceed, and for supporting our conclusions. We agree that the FEFI results regarding deformation and sliding are unrealistic, and we certainly do not want our study to be cited as evidence that "SIA-like flow can be expected at the Thwaites Glacier grounding line." However, due to the lack of direct observations in the region, and considering that other studies (e.g., McCormack et al. (2022)) indicate the deformational regime is at least heterogeneous, we were hesitant to entirely rule out deformational flow at the TG grounding line based on physical intuition alone. Without sufficient (observational) evidence, such exclusion would also be not justified in our opinion. Still, there is value in exploring FEFI as a what-if scenario that cannot be completely ruled out, as it is common for models using data assimilation as an initialization technique to alter their flow enhancement factor. The features that we found could be present in those models as well.

We want to make it clear that our goal is not to produce the best possible inversion outcome, nor do we want to deliver a state-of-the-art projection (e.g., we do not use a realistic climate forcing). Instead, our aim is to demonstrate that two equally valid inversions (with different advantages and weaknesses), which result in similar present-day ice sheets, can lead to very different responses in an unforced future simulation.

Regarding your option 1) (Remove FEFI from manuscript except some side notes), we cannot maintain our current results and conclusions - which we're pleased the reviewer supports - if we rely solely on DI simulations. The FEFI simulations are essential to our analysis. Hence, we would like to keep those in the main text rather than moving them entirely to the supplementary, which in fact may still lead to a deliberate misinterpretation. As for option 2) (Argue that FEFI results are credible), we agree that this would be a substantial undertaking, and we would prefer to avoid it due to the risk of the paper being misinterpreted as evidence that TG exhibits SIA-like behaviour. Finally, regarding option 3) (Improve the FEFI procedure), we agree this would be an excellent direction for a future

study, including further improvements to the FEFI method, such as possibly advecting ice properties along flow lines. However, it is indeed not realistic to include this in this manuscript.

Therefore, we agree with the reviewer that we should clearly state that we do not interpret the deformation-dominated ice flow for Thwaites Glacier (TG) (or the inverted flow enhancement factor more broadly) as a physical property of the ice sheet. Instead, we view $E$ as a tunable parameter used to better match present-day conditions, which simultaneously increases the underdetermined nature of the inverse problem. The purpose of introducing this new inversion, alongside the DI approach, is to generate an alternative solution to demonstrate that the choice of initialization method is the primary driver of divergent future projections in our simulations—and to offer a physical explanation for why this is the case. Following this line of thought, we will change our text.

We will rewrite the following paragraph L376-L386 with the following:
'This is particularly striking at the Western TG grounding line, where at present the regionally highest ice surface velocities are observed, which are unlikely to be solely by deformation. Sliding is expected to be the dominant regime of a fast-flowing (Antarctic) outlet glacier from standard ice flow theory, and the SSA is widely used as the appropriate stress approximation to model these regions (e.g. Bueler and Brown (2009); Brondex et al. (2019); Gudmundsson et al. (2023); Morlighem et al. (2024)). However, McCormack et al. (2022) modelled the ice flow regime more extensively and possibly more physically than we do, and found a heterogenous pattern of sliding and deformation close to the TG grounding line depending on the flow law used. Therefore, a mix of sliding and deformation cannot be excluded entirely (see Fig 2 in Mccormack et al. (2022)).

Because our model uses Glen's flow law (which was developed for isotropic, secondary creep) it cannot capture e.g. tertiary creep and ice damage accurately (Glen, 1952; Budd et al., 2013; Graham et al., 2018). By inverting viscosity, we are effectively compensating for this missing process, so the resulting flow-enhancement factor and inferred flow regime reflect model deficiencies rather than intrinsic ice properties. This is particularly true in regions with fast-flowing ice, because our FEFI initialization is only allowed to change the flow enhancement factor in areas where the modelled ice surface velocity errors exceed 25 m yr$^{-1}$. In addition, FEFI assigns ice properties to fixed grid cells instead of advecting them along flowlines, even though impurities, damage, and fabric anisotropy are fundamentally Lagrangian properties of the material. If the ice is damaged, it will remain so downstream of where the damage was initiated. The FEFI inversion can therefore generate physically questionable enhancement factors that mask upstream errors in the flow regime. For these reasons, we doubt that deformation dominates at the TG grounding line, since the current inversion cannot yet reproduce physically consistent ice properties. Observations could provide clarity on the flow regime of the TG grounding line. In particular, measurements in the vertical structure of the horizontal velocity in critical regions will help to distinguish which flow regime dominates.

**Specific Comments**

Abstract: obviously, if the paper is revised as I suggest, the last few lines are only weakly supported and so should not appear in the abstract.

We agree, and as replied to reviewer 1, we will remove 'The latter makes it difficult to base general claims on ice sheet modelling results'

L76 T = \beta u = … . The \beta u is not needed – many models do this as part of their implementation, but don't think \beta is mentioned again.

We will remove \beta_u

L95. Eq 1.4 is sometimes called a Budd law.

We will add to L93 '..and is referred to as the 'Budd' law after Budd et al. (1979)'

L114. In all four laws friction increases with speed, with diminishing returns (but tend to a limit in the ZI/Schoof cases)

We will add to L114, '..with diminishing returns.'

Figure 1. The asymptotes could be added to the figure.

Great suggestion, we will add the asymptotes (e.g. Cc*N) to the figure.

Eq 1.7 R_f rather than \Chi _f ? \Chi is dimensionless, but \Chi f is a stress (like the R components)

We agree that this avoids confusion and will change Chi_f to R_f in Eq 1.7 and 1.8

L141. \Chi is not a term

We will rewrite L141 to 'The variable \Chi in Eq 1.7 has a value of 0 for areas that are unbuttressed: the buttressing force then equals the driving stress if the ice sheet ended at that point with an ice cliff'

 L145. Sorry, I don't see the logic here.

We will remove 'The former assumption implies that high basal friction just upstream of the grounding line, which will oppose ice flow, is interpreted as buttressing as well'

L148. 'Shelf kill'. In the interests of a less macho phrase, how about 'Shelf removal'. I know that shelf kill has been used elsewhere.

Great suggestion, we will replace 'Shelf kill' with 'Shelf removal' throughout the manuscript

L155. Purest -> simplest / most direct?

We will replace 'Purest' with 'simplest'

L160 – the whole paragraph refers to something in the supplement, but I don't think you need to further show the utility of these well-know buttressing indicators.

We would like to show that the buttressing indicators do not necessarily correlate for complex geometries such as the AIS. We will add to L163: 'little to no correlation could be found when the same method was applied to AIS ice shelves, as shown in Fig S9.'

Section 2.3 – this section seems a little disorganised. In particular, the nudging equations are introduced immediately before a general introduction to the nudging approach. The tables could be moved to an appendix since the parameters are usually defined in-text.

We will move Table 1 and Table 2 to the supplementary material. We will move L217-224 to L181, to start with a description of the nudging before introducing the equations.

L227 – I would like more detail (i.e math expressions) at this point.

We will add a mathematical expression as follows below L230:

$$\frac{dC_c}{dt} > 0 \;\wedge\; \frac{dE}{dt} < 0 \;.$$
(1.21)

We will add to L229: (e.g. when modelled ice is too slow and too thin, the basal friction inversion will try to increase the ice thickness by decreasing ice velocities, and the flow enhancement factor inversion will try to increase the ice surface velocities, mathematically shown in Eq 1.21).

L265 – 'other factors' – e.g damage, fabric formation, errors in the temperature field

We will add '(e.g., damage, fabric formation and (local) anisotropy and errors in the temperature field)'

Fig 4 – could the panels be larger/split up?

We will include zooms here and larger figures in the supplementary materials. We will add the inverted ocean temperature perturbation of the whole ice sheet to the supplementary material, and refer to that in L332 and L353. We will include zoom-ins of the velocity misfit in the ASE region, including a discussion on the grounding line flux as mentioned in response to one of the major comments above.

Fig 10 – actually a more general comment – the shelf removal figures are the more useful in your text, whereas the buttressing number only helps you (vaguely) reiterate a known point about the two halves of Thwaites ice shelf / buttressing being greater near the GL. I would remove the buttressing number analysis, so that the right-hand panels of fig 10 can be larger.

We agree with the reviewer that our buttressing analysis with the buttressing number in the current version of the manuscript does not add new insights. We will reformulate our results somewhat in response also to comments from reviewer 1, to use the buttressing number solely for the purpose of quantifying ice shelf strength, and the acceleration number to quantify the basal friction strength. Then the buttressing number is explicitly used in the text to demonstrate differences in buttressing strength. We would then keep Fig 10 as is but rewrite L465-472:

'The right side of Figure 10 shows changes in ice velocity during the shelf-removal experiments at years 250 and 500, comparing Zoet-Iverson sliding (left column) and power law sliding (right column). Following the removal of the ice shelves, grounded ice in the Zoet-Iverson simulation accelerates rapidly—especially at the Thwaites Glacier (TG) grounding line and farther inland—much more so than in the power law case. Two primary mechanisms can slow down a retreating marine-terminating glacier like TG: buttressing and basal friction. At year 500, buttressing at the TG grounding line is stronger in the Zoet-Iverson simulation than in the power law case, as indicated by the higher buttressing numbers. Despite this, the acceleration response to shelf loss (right four panels of Fig. 10) is greater in the Zoet-Iverson case. This is because, in the power law case, basal friction increases with velocity (as shown in Fig. 1), limiting the glacier's speed-up. In contrast, the Zoet-Iverson simulation exhibits less frictional resistance and thus stronger acceleration. We conclude that, during the TG collapse, buttressing is the primary braking mechanism in the Zoet-Iverson case, whereas increased basal friction dominates in the power law case. Interestingly, due to the specific bed geometry of TG and the conditions in the unforced simulation, both scenarios produce a similar contribution to global mean sea level rise—though driven by different mechanisms. '

Fig 11. If you *do* include the FEFI results (I suggest not), then show both DI and FEFI at years 575 and 775 (i.e. four panels).

We will add two panels showing the ice thickness difference in DI at 775 years, and FEFI ice thickness changes at 575 years.

L580- 600 – clearly would be removed if you remove the FEFI material.
We would not like to remove the FEFI material, as stated above to the major comment, so we would like to keep this section in.

L602 – you could compare your E(x, y) with other model results.

We will add:

'Our results generally align with the heterogeneous pattern of deformational flow reported by McCormack et al. (2022) with the notable exception of a localized patch of deformation-driven flow at the Thwaites Glacier grounding line in our simulations. Barnes et al. (2021) examined the transferability of inverted parameters across three ice sheet models and found substantial variation in the inverted rate factor among them. Although a direct comparison with our inverted flow enhancement factor is difficult, since the rate factor also depends on temperature (see Eq. 1.16), the inverted rate factors in the Úa model (see Fig. 4 in Barnes et al. (2021)) vary by up to two orders of magnitude. This is consistent with the heterogeneous patterns we observe in our own results. Similarly, Brondex et al. (2019) show vertically averaged viscosity patterns with lower values near the grounding line compared to inland ice. This may be due to their assumption of applying only the SSA. While their results appear more homogeneous than those in Barnes et al. (2021), spatial variability is still present.'

L615 – I don't agree in this case because the other models mentioned don't differ from one another in the same way as the DI and FEFI models differ from one another. I am sure you are correct to say that differing initial conditions could explain any number of discrepancies between models, I just do not think the FEFI/DI contrast is representative.

We will rephrase L615 to: The results presented in this study illustrate why CISM can be initialized with more or less sensitivity to the choice of basal sliding law. Carrying out these types of experiments with other ice sheet models will enhance our understanding of why some simulations (Brondex et al., 2017; Sun et al., 2020; Brondex et al., 2019), are more sensitive to changes in basal friction laws than others (Barnes and Gudmundsson, 2022; Wernecke et al., 2022), and possibly lead to similar conclusions.

**Technical Corrections**

Eq1. Use f(x, y) notation to make spatially varying vs constant parameters clear?

We will add $C_c(x, y)$ to show the spatial variability of Coulomb C.

80 $\tan \phi$, not $tan \phi$. In a similar vein, there are frequent italic subscripts in equations that should probably be roman (e.g in eqns 1 & 2)

We will replace those with $\tan \phi$ and we will remove italic subscripts in our equations

L112 'asymptote' is not a verb (usually).

We will replace 'asymptote' with 'approaches a limit'

**References**

Barnes, J. M. and Gudmundsson, G. H.: The predictive power of ice sheet models and the regional sensitivity of ice loss to basal sliding parameterisations: a case study of Pine Island and Thwaites glaciers, West Antarctica, The Cryosphere, 16, 4291-4304, 2022.

Barnes, J. M., Dias dos Santos, T., Goldberg, D., Gudmundsson, G. H., Morlighem, M., and De Rydt, J.: The transferability of adjoint inversion products between different ice flow models, The Cryosphere, 15, 1975-2000, 2021.

Brondex, J., Gillet-Chaulet, F., and Gagliardini, O.: Sensitivity of centennial mass loss projections of the Amundsen basin to the friction law, The Cryosphere, 13, 177-195, 10.5194/tc-13-177-2019, 2019.

Brondex, J., Gagliardini, O., Gillet-Chaulet, F., and Durand, G.: Sensitivity of grounding line dynamics to the choice of the friction law, Journal of Glaciology, 63, 854-866, 2017.

Budd, W., Keage, P., and Blundy, N.: Empirical studies of ice sliding, Journal of glaciology, 23, 157-170, 1979.

Budd, W. F., Warner, R. C., Jacka, T., Li, J., and Treverrow, A.: Ice flow relations for stress and strain-rate components from combined shear and compression laboratory experiments, Journal of Glaciology, 59, 374-392, 2013.

Bueler, E. and Brown, J.: Shallow shelf approximation as a "sliding law" in a thermomechanically coupled ice sheet model, Journal of Geophysical Research: Earth Surface, 114, 2009.

Glen, J.: Experiments on the deformation of ice, Journal of Glaciology, 2, 111-114, 1952.

Graham, F. S., Morlighem, M., Warner, R. C., and Treverrow, A.: Implementing an empirical scalar constitutive relation for ice with flow-induced polycrystalline anisotropy in large-scale ice sheet models, The Cryosphere, 12, 1047-1067, 10.5194/tc-12-1047-2018, 2018.

Gudmundsson, G. H., Barnes, J. M., Goldberg, D., and Morlighem, M.: Limited impact of Thwaites Ice Shelf on future ice loss from Antarctica, Geophysical Research Letters, 50, e2023GL102880, 2023.

McCormack, F., Warner, R., Seroussi, H., Dow, C., Roberts, J., and Treverrow, A.: Modeling the deformation regime of Thwaites Glacier, West Antarctica, using a simple flow relation for ice anisotropy (ESTAR), Journal of Geophysical Research: Earth Surface, 127, e2021JF006332, 2022.

Morlighem, M., Goldberg, D., Barnes, J. M., Bassis, J. N., Benn, D. I., Crawford, A. J., Gudmundsson, G. H., and Seroussi, H.: The West Antarctic Ice Sheet may not be vulnerable to marine ice cliff instability during the 21st century, Science Advances, 10, eado7794, 2024.

Sun, S., Pattyn, F., Simon, E. G., Albrecht, T., Cornford, S., Calov, R., Dumas, C., Gillet-Chaulet, F., Goelzer, H., and Golledge, N. R.: Antarctic ice sheet response to sudden and sustained ice-shelf collapse (ABUMIP), Journal of Glaciology, 66, 891-904, 2020.

Wernecke, A., Edwards, T. L., Holden, P. B., Edwards, N. R., and Cornford, S. L.: Quantifying the impact of bedrock topography uncertainty in Pine Island Glacier projections for this century, Geophysical Research Letters, 49, e2021GL096589, 2022.

---

## Author Response (AR4)

**Author replies reviewer #2**

Author replies are in blue

I appreciate the authors' efforts in responding to the previous comments and in improving the clarity of the results section, which is now much easier to follow. I also understand that, following the removal of the FEFI initialisation, the authors sought an alternative initial state that would allow them to preserve the key messages of the study.

However, I am puzzled by the current comparison between the simulations starting from the P05 and P1 initial states for the power law case. Because the power law is independent of N, and therefore of p, the P05 and P1 initialisations should theoretically produce identical initial states and thus identical forward simulations. The differences presented here stem entirely from the fact that the initial states were first generated using the Zoet–Iverson law (with p = 0.5 and p = 1), and only then rescaled to a power-law-equivalent friction field. While I understand that the authors aim to assess the influence of initialisation choices on the friction field and subsequent simulations, I am not yet convinced that this comparison is meaningful. I believe this point needs to be discussed and more clearly justified in the manuscript.

Once this issue is addressed, together with the specific comments listed below, I consider the manuscript suitable for publication in The Cryosphere.

We thank the reviewer again for their thorough look through our manuscript and agree that it needs to be better explained that we do two initializations, both with the ZI and start the powerlaw simulations from them by rewriting the free parameter in the powerlaw. This indeed implies that the P1 and P05 simulations done with the powerlaw differ because of their slightly altered initial state acquired through an initialization with the ZI law. We would also like to stress that these two distinct initializations and their subsequent evolutions are not intended to produce the most accurate possible future projections. Rather, they are designed to generate two physically plausible ice-sheet evolutions, plausible in the sense that, if the bedrock state matches current estimates, the simulated evolution is as realistic as current models allow. Together, they demonstrate that ice-shelf buttressing can, but does not necessarily, mitigate the influence of the basal sliding law on ice-sheet evolution.

We will add to Ln 282: 'Three of the friction laws evaluated in this study depend on the effective pressure, whereas one (the powerlaw) does not. As a result, any differences between continuation experiments initialized from either P1 or P05 using the powerlaw friction law arise solely from small variations in the initialized friction field, obtained with the ZI law. During a continuation experiment, the free parameter $C_p$ can be regarded as analogous to the product of $C_c$ and the effective pressure $N$ in the other three friction

laws. While the product $C_c N$ evolves differently for different values of $p$, $C_p$ remains unchanged.

We will also add to Ln 389: As shown in Fig. 8c, a distinct band near the TG grounding line exhibits higher friction under the P1 initialization than under P05, whereas the opposite pattern occurs near the PIG grounding line. This indicates that, when P1 is chosen, CISM tends to strongly stabilize TG while destabilizing PIG. This contrast has a profound impact on the sequence of collapse observed in the continuation experiments discussed in the next section, even for the effective-pressure-independent power-law friction law.

Specific comments:

l.12–13: I suggest rephrasing as:
"We find a geometry-driven connection between buttressing and basal sliding in the Amundsen Sea Embayment when performing multi-century future simulations based on the present-day observed imbalance of the Antarctic Ice Sheet, in which Thwaites and Pine Island glaciers eventually collapse."

This is a great suggestion, we will replace lines 12-13 with the suggested sentence.

l.23–24: Isn't it the present-day imbalance rather than the present-day ocean thermal forcing that drives the collapse?

Yes it is. However, including the present-day imbalance has an influence on the initialized present-day ocean thermal forcing: when using the present-day imbalance CISM needs less negative temperature perturbations in our treatment of the basal melt calculation compared to an initialization without the imbalance. Therefore, we argue that our 'imbalance' initialization produces ocean conditions better in line with the observations than our equilibrium initialization. In turn, those different initialized ocean conditions then produce the imbalance in a future simulations. We will add 'the present-day ocean thermal forcing, which is a product of the inversion using the present-day imbalance'

l.71–72: What is meant by 'for realistic collapse conditions'? Please clarify.

We meant compared to schematic or schematically-forced experiments (i.e. uniform warming in the ASE). We will replace 'for realistic collapse conditions' with 'for sustained present-day forcing'.

l.118: Replace the period with a comma in: "the bedrock height, and p a constant in the range …"

We will do this

l.167–169: The sentence about applying friction and/or basal melt scaled by the grounded fraction of a grid cell (PMP; Leguy et al., 2021) seems redundant, as this is already described in lines 219–221.

We will remove this sentence here

l.162–164: It should be explicitly stated here that p has no influence in the power-law formulation, since it does not depend on N (Eq. 1.2).

We will add to Ln 164 (Ln 139 in file uploaded in the TC upload system, due to some confusions with the manuscript versions): 'Note that the powerlaw does not depend on the effective pressure'

l.249: "DI" is no longer relevant and should be replaced with "P05."

We will replace 'DI' with 'P05'

l.246, Section 2.4.1: Again, it should be clarified in this section that the P1 initialisation will be similar to P05 for a power law.

We will add to Ln 230: 'These two initializations will serve as a starting point for continuation simulations done with the four different friction laws described above to produce a set of 8 forward simulations. We will rewrite free parameters in the friction laws to be able to start every forward simulations from a single initialization simulation.'

l.282: "We take the initialisations using the ZI law as initial states". Ok, I understand better now. I think that it is very important to explicitly acknowledge that your results are entirely dependent on this choice. If the power law had been used for initialisation, the P05 and P1 initial states would be identical. The fact that starting from the ZI law introduces differences between the P05 and P1 power-law simulations requires a much stronger justification.

We agree with the reviewer that this is not clear in our manuscript, and we hope that the addition to Ln 230 (previous comment), and Ln 389 and Ln 282 as our reply to the main comment will clarify this in the manuscript.

l.340–348: This discussion appears to refer to an initialisation performed using observed ice velocities. Since the current version of the manuscript no longer uses observed velocities in the initialisation, it is unclear what message is intended here. As written, this paragraph no longer seems relevant.

We will remove the parts about the ice surface velocities from this paragraph.

l.596: Again, I would say that it is the present-day imbalance accounted for in the initialisation that leads to the collapse of both glaciers under the present-day ocean forcing; not simply applying the present-day ocean forcing.

We will add 'applying the inverted present-day ocean forcing' to emphasize the effect of including the present day mass change rates on the inverted ocean temperatures.